# High-fidelity parametric beamsplitting with a parity-protected converter

Yao Lu [1,2,3] ✉, Aniket Maiti [1,2,3] ✉, John W. O. Garmon[1,2], Suhas Ganjam[1,2], Yaxing Zhang[1,2], Jahan Claes [1,2], Luigi Frunzio [1,2], Steven M. Girvin [1,2] & Robert J. Schoelkopf[1,2] ✉

Fast, high-fidelity operations between microwave resonators are an important tool for bosonic quantum computation and simulation with superconducting circuits. An attractive approach for implementing these operations is to couple these resonators via a nonlinear converter and actuate parametric processes with RF drives. It can be challenging to make these processes simultaneously fast and high fidelity, since this requires introducing strong drives without activating parasitic processes or introducing additional decoherence channels. We show that in addition to a careful management of drive frequencies and the spectrum of environmental noise, leveraging the inbuilt symmetries of the converter Hamiltonian can suppress unwanted nonlinear interactions, preventing converter-induced decoherence. We demonstrate these principles using a differentially-driven DC-SQUID as our converter, coupled to two high-Q microwave cavities. Using this architecture, we engineer a highly-coherent beamsplitter and fast (~100 ns) swaps between the cavities, limited primarily by their intrinsic single-photon loss. We characterize this beamsplitter in the cavities' joint single-photon subspace, and show that we can detect and post-select photon loss events to achieve a beamsplitter gate fidelity exceeding 99.98%, which to our knowledge far surpasses the current state of the art.

The precise manipulation of high-Q bosonic modes is vital to experimentally explore a wide range of phenomena in many-body physics and quantum information. A crucial component in this endeavor is a programmable two-mode interaction, exemplified by the time-dependent beamsplitter Hamiltonian $\hat{\mathcal{H}}_{BS}/\hbar = g_{BS}(t)(e^{i\varphi_{BS}}\hat{a}^\dagger\hat{b} + e^{-i\varphi_{BS}}\hat{a}\hat{b}^\dagger)$. This controlled photon-exchange coupling is a requirement for continuous variable quantum computation[1], and has direct applications in bosonic simulations of hopping models and lattice gauge theories[2–5]. It is particularly appealing to implement such a Hamiltonian in circuit-QED, a flexible platform offering readily available nonlinear control of high-Q modes in superconducting resonators[6]. These resonators primarily lose coherence through single-photon loss[7,8], a 'noise-bias' that has been utilized in important demonstrations including bosonic error-correction[9–12] beyond break-even[13–15].

However, the experimental implementation of a fast, high-fidelity beamsplitter that preserves the long lifetime of these resonators and does not introduce additional decoherence has remained a challenge. If realized, this interaction would help construct logical entangling gates between qubits encoded in the resonators[16–18], and enhance long-distance interactions through microwave quantum buses[19,20].

Within the circuit-QED framework, interactions between linear superconducting resonators can be implemented by coupling them via a Josephson junction-based nonlinear 'converter'[21–23]. Driving this converter with RF drives can actuate a 'parametric beamsplitter'[24–29] between modes that are widely separated in frequency space, offering large on-off ratios. The amplitude of the drives sets the strength of the beamsplitter, so one could ideally improve both the speed and fidelity of beamsplitting-based gates by simply driving harder. However,

[1]Departments of Applied Physics and Physics, Yale University, New Haven 06511 CT, USA. [2]Yale Quantum Institute, Yale University, New Haven 06511 CT, USA. [3]These authors contributed equally: Yao Lu, Aniket Maiti. ✉e-mail: physics.lu@yale.edu; aniket.maiti@yale.edu; robert.schoelkopf@yale.edu

controlling the dynamics of strongly driven, dissipative, nonlinear systems can be challenging[30,31]. The wide-bandwidth of the Josephson nonlinearity can activate numerous parasitic processes that cause exchanges with modes lossier than the resonators, including with the converter itself, spoiling fidelity. The converter can also be incoherently excited through a dressing of its natural decoherence in the presence of the drives, which can directly dephase the beamsplitter interaction and harm the noise-bias of the resonators. These parasitic processes and drive-induced coupler excitations have been significant barriers in previous implementations[16,26] and suppressing them is crucial to engineering a clean, high-fidelity beamsplitter.

One approach that has been used to tame this nonlinearity is engineering multi-junction converters with useful symmetries[21,32], that prevent a significant fraction of the nonlinear processes allowed in a single-junction circuit like the transmon[33]. In this work we employ a similar approach, using the familiar circuit of the symmetric DC Superconducting QUantum Interference Device (SQUID), with careful engineering to make it compatible with a high-Q environment. Making full use of this symmetry requires the SQUID to be driven in a purely differential manner, and we introduce an architecture for delivering this drive through an auxiliary 'buffer' mode.

In this work, we use the differentially-driven SQUID (DDS) to perform a fast, highly coherent beamsplitter between two high-Q 3D superconducting cavity resonators. We show that the coupler experiences almost no drive-induced excitation, independent of the beamsplitter rate. The beamsplitter fidelity is then limited only by the cavities' single-photon decay, thus being highly compatible with existing bosonic encodings[32,34,35]. As a first demonstration, we characterize the pulsed operation of this beamsplitter with single photons in a microwave implementation of the dual-rail qubit subspace $\{|0_a 1_b\rangle, |1_a 0_b\rangle\}$, where the beamsplitter interaction provides universal control. We achieve an average gate fidelity of 99.92%, which on detecting single-photon loss events is boosted to 99.98%. This paves the way for quantum computing architectures based on erasure-limited dual-rail qubits[12,24,36], and for cleaner parametric processes in general.

## Results

### Engineering a differentially-driven SQUID

We now outline the key insight of properly leveraging the symmetry of the SQUID for a cleaner beamsplitter, and describe an architecture that accomplishes it. The SQUID has two orthogonal modes, a common 'coupler' mode with symmetric Josephson phases across its two junctions, and a differential 'actuator' mode with anti-symmetric phases. Fully utilizing the symmetry of these modes requires a precisely-engineered selective coupling of the coupler and actuator to the resonators and the drive, respectively (Fig. 1a), at the sweet-spot of zero DC flux. This is different from a conventional flux-driven SQUID[37–40], since extra care must be taken to ensure that the drive does not excite the common mode. This drive must also be introduced in a minimally invasive manner that does not spoil the lifetime of either the resonators or the coupler.

The Hamiltonian of such a differentially-driven SQUID is (Supplementary Note 1):

$$\hat{\mathcal{H}}_{\text{DDS}} = 4E_C \hat{n}_c^2 - E_J \underbrace{\cos(\phi_d)\cos(\hat{\theta}_c)}_{\text{Even Parity}}, \tag{1}$$

where $\hat{\theta}_c = (\hat{\theta}_1 + \hat{\theta}_2)/2$ and $n_c$ are the conjugate phase and charge variables describing the coupler mode, and $\phi_d = \langle \hat{\theta}_1 - \hat{\theta}_2 \rangle/2$ is the classical response of the driven actuator, for junction phases $\theta_{1,2}$. $E_J$ and $E_C$ are the total Josephson and charging energies, respectively. The advantage of engineering this Hamiltonian is two-fold. First, the Hamiltonian is protected from processes that involve an odd number of mode quanta, while preserving the strength of desired quadratic process that provides the beamsplitting interaction. This 'parity protection' forbids up to half the parasitic coherent processes allowed by a single-junction converter, including the swapping of sensitive information into the coupler that limited previous implementations[26]. Second, driving through the orthogonal actuator port provides flexibility in choosing the drive frequencies, which we can utilize to drive far away from non-protected processes that could otherwise limit fidelity. Combining

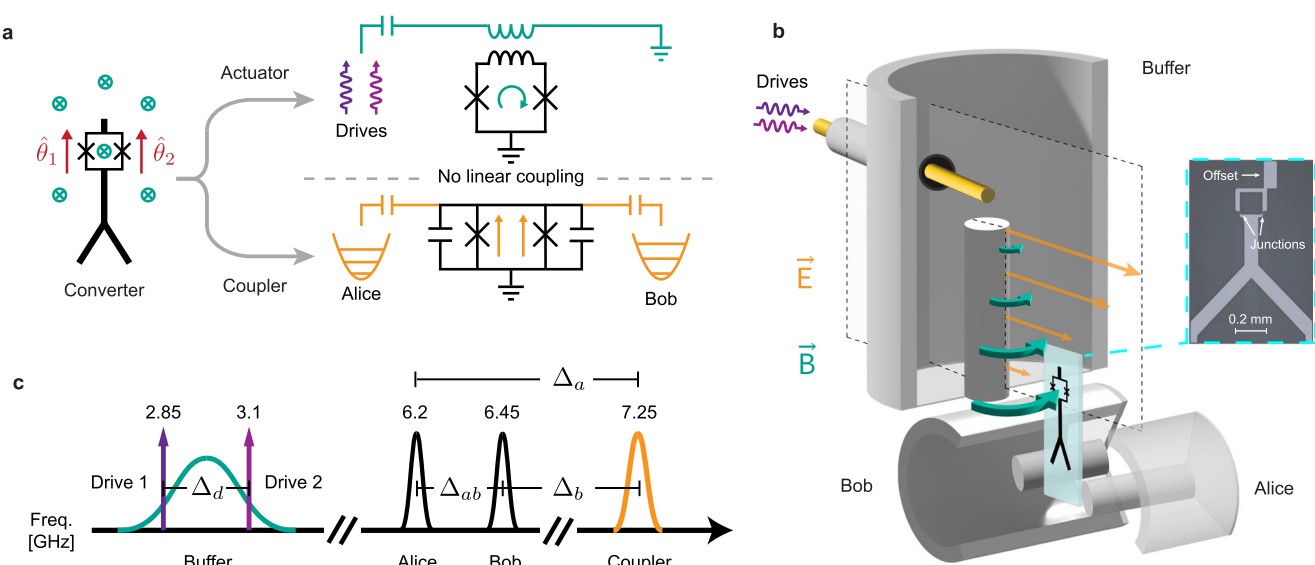

**Fig. 1 | The differentially driven SQUID as a parity-protected converter. a** The symmetric DC-SQUID contains two orthogonal modes[57,58], the common mode (coupler) and the differential mode (actuator). We selectively couple the former to two bosonic modes and the latter to the drives to take advantage of the natural symmetries of the Hamiltonian in Eq. (1). **b** Implementing the purely differential drive through a 3D buffer post-cavity (figure is exaggerated for illustrative purposes). The natural separation in electric and magnetic fields in the $\lambda/4$ mode is used to purely drive the actuator, without exciting the coupler. The sensitive

quantum information is stored in two high-Q $\lambda/4$ post-cavities (Alice and Bob) that participate in the coupler, enabling parametric beamsplitting between them. The inset shows an optical micrograph of the SQUID device, displaying the purposely offset antenna pad that counters residual drive-asymmetry. **c** Frequency stack for relevant modes in the system. The difference of the two drive frequencies ($\Delta_d$) is fixed to be equal to the cavity detuning ($\Delta_{ab}$) for resonant beamsplitting. The drives are placed symmetrically around the buffer mode resonance, which is engineered to be far-red detuned from the coupler frequency.

these two advantages has the potential to allow the suppression of not only unwanted coherent processes, but also incoherent coupler excitations[31,41] due to coupler decoherence dressed by the drives (see Supplementary Note 2 for a detailed description).

Implementing the symmetric Hamiltonian in Eq. (1) amounts to fabricating a SQUID with symmetric junctions, calibrating it to zero DC flux, and delivering a purely differential drive. The first two conditions are possible to optimize in fabrication or calibrate out, respectively, and have been described in Supplementary Note 7. Engineering the differential drive, however, necessitates the delivery high-frequency flux in a superconducting package, while simultaneously eliminating stray electric fields that couple to the common mode of the SQUID and accounting for spatial gradients in the time-dependent flux (see Methods "Fine-tuning a differential drive" for more details).

We simultaneously fulfill these constraints using the $\lambda/4$ mode of a stub cavity (dubbed the 'buffer mode'), integrated into the same monolithic package that hosts the high-Q storage modes (Fig. 1b). Once driven, the buffer mode provides an oscillating magnetic field that serves as the flux drive penetrating the SQUID loop. We place the SQUID at the base of the buffer mode, which functions as a virtual ground, minimizing the common mode drive due to the driven electric field while maximizing the flux penetrating the SQUID loop. The coupling to any remnant electric field is further suppressed by orienting the SQUID's electric dipole moment to be perpendicular to the driven field. In addition to this minimization of the common-mode drive, driving through the buffer mode also has the benefit of imposing a finite bandwidth (~250 MHz) on the noise spectrum of the environment seen through the drive line.

Finally, the full elimination of the common mode drive is achieved by a deliberate offset of the SQUID's capacitive pad. This counters the effects of flux gradients in our system, by engineering an additional flux-dependent electromotive force, which can be directly optimized using finite-element-method simulations (see Methods). In experiment, imperfections in fabrication or placement of the SQUID chip might affect the achievable drive orthogonality. However, we are able to directly estimate this orthogonality in our actual device by experimentally comparing the strength of an allowed process to one that should be forbidden by parity protection, finding a residual common drive on the order of $|\phi_c/\phi_d| \sim 1\%$ (see Methods "Experimentally characterizing residual drive asymmetry").

### Demonstrating a high-coherence beamsplitter

We now present the full experimental realization (Fig. 1b) of a high-fidelity beamsplitter that strongly suppresses undesirable coupler heating. Our construction consists of a high-purity aluminum package hosting three coaxial $\lambda/4$ stub cavity modes: the buffer mode and the two high-Q storage modes[6]. The storage modes are capacitively coupled to the Y-shaped antenna of the SQUID[42] and have a negligible mutual inductance to the SQUID loop, ensuring that they exclusively participate[43,44] in the coupler mode:

$$\hat{\theta}_c \approx \left(\frac{2E_C}{E_J}\right)^{\frac{1}{4}}\left(\frac{g_a}{\Delta_a}\,\hat{a} + \frac{g_b}{\Delta_b}\,\hat{b} + \hat{c}\right) + \text{h.c.} \tag{2}$$

Here $\hat{a}$, $\hat{b}$ are the ladder operators for our dressed storage modes, named Alice and Bob, respectively, while $\hat{c}$ represents the dressed coupler. The coupling strengths ($g_{a,b}$) and mode detunings ($\Delta_{a,b}$, Fig. 1c) between the storage modes and the coupler are chosen to be in the dispersive regime ($\frac{g_a}{\Delta_a}, \frac{g_b}{\Delta_b} \sim 0.1$). In addition, we can prepare and readout Fock states in Bob through a dispersively coupled ancilla transmon[45] and a dedicated stripline readout resonator. The SQUID coupler also includes a dedicated readout resonator and drive pin, for explicit characterization of frequency shifts and drive-induced excitation. The measured device parameters are presented in Supplementary Note 12, and details of device fabrication can be found in Supplementary Note 11.

We activate and control the amplitude and phase of our beamsplitter with a bi-chromatic drive on the actuator: $\phi_{d_{1,2}}(t) = |\phi_{d_{1,2}}|\cos(\omega_{d_{1,2}}t + \varphi_{d_{1,2}})$. When the difference in our drive frequencies ($\Delta_d$) is close to our cavity detuning ($\Delta_{ab}$), Eqs. (1) and (2) combine to create a tunable beamsplitter Hamiltonian (see "Methods" for full derivation):

$$\hat{\mathcal{H}}_{\text{BS}}/\hbar = \Delta_{\text{BS}}\,\hat{a}^\dagger\hat{a} + g_{\text{BS}}\left(e^{i\varphi_{\text{BS}}}\hat{a}^\dagger\hat{b} + e^{-i\varphi_{\text{BS}}}\hat{a}\hat{b}^\dagger\right), \tag{3}$$

$$\text{with}\quad g_{\text{BS}} \approx \frac{\omega_c}{2}\frac{g_a g_b}{\Delta_a \Delta_b}J_1\left(|\phi_{d_1}|\right)J_1\left(|\phi_{d_2}|\right), \tag{4}$$

$$\Delta_{\text{BS}} = \Delta_{ab} - \Delta_d + \Delta_{Z,ab} \tag{5}$$

where $J_1(|\phi_{d_{1,2}}|)$ is the first-order Bessel function of the drive amplitudes, and $\varphi_{\text{BS}}$ is the beamsplitter phase, controlled by the relative phase of the drives. The drives also induce an additional frequency offset, the relative AC-Zeeman shift of the cavities, $\Delta_{Z,ab}$, and we experimentally find the amplitude-dependent condition $\Delta_{\text{BS}} = 0$ to execute a resonant beamsplitter.

We characterize our beamsplitter interaction using the joint single-photon subspace of our storage cavities, which forms a microwave implementation of a dual-rail qubit (Fig. 2a). We initialize a single photon in Bob with a preparation fidelity of ~94%, by displacing Bob to a coherent state ($\alpha_b = \sqrt{2}$) and using number-resolved measurements through the ancilla to post-select the desired Fock state. We then apply the resonant beamsplitter interaction for a range of times up to 32 μs (Fig. 2b), to estimate both the beamsplitting strength and the driven decoherence time of the dual-rail qubit. The evolution of the time-dependent photon population in Bob follows (see Methods "Decoherence in the dual-rail subspace" for derivation):

$$P_{\text{Bob}} = \frac{1}{2}e^{-\kappa_1 t}\left(1 + e^{-\kappa_\varphi t}\cos(2g_{\text{BS}}t)\right) \tag{6}$$

Here, $\kappa_1$ is the mean of the driven cavities' single-photon decay rates, and is the effective rate of population leakage out of the dual-rail subspace into vacuum $|0_a 0_b\rangle$. The dual-rail qubit may also experience dephasing at a rate $\kappa_\varphi$, which would drive this evolution towards an evenly mixed state within the qubit subspace. Combining these lets us place a lower bound on the expected decoherence limit on the fidelity of a single beamsplitter operation (see "Methods"):

$$\mathcal{F} \approx 1 - \frac{\pi}{4}\frac{\kappa_{\text{BS}}}{g_{\text{BS}}}, \quad \kappa_{\text{BS}} = \kappa_1 + \frac{\kappa_\varphi}{2}, \tag{7}$$

which is a more accurate metric than the one used in ref. 26.

We choose our operating point to maximize this expected fidelity $\mathcal{F}$ with respect to drive strength, which we characterize by analyzing sections of the resulting driven long-time evolution at various drive amplitudes (Fig. 2c). At each amplitude, we extract the $g_{\text{BS}}$ and $\kappa_{\text{BS}}$ by fitting two short sections of the evolution (Methods) to Eq. (6). We are able to obtain a maximum $g_{\text{BS}}/2\pi$ exceeding 5 MHz, with the effective decoherence-limited fidelity surpassing 99.9% for a wide range of beamsplitting strengths. We observe that upon increasing the drive strength, the coupler frequency shifts closer to the cavities (by ~200 MHz), which may be accompanied by direct sideband interactions between the coupler and the cavities (Supplementary Note 5), leading to an increased hybridization between the modes. This effect results in a faster-than-quadratic dependence of the beamsplitting strength on our drive amplitudes, but limits the fidelity of the beamsplitter at higher amplitudes due to the aggravated Purcell loss.

At the operating point, we fit the evolution in Fig. 2b to find $g_{\text{BS}}/2\pi = 2.16 \pm 0.01$ MHz, $\kappa_1 = (197 \pm 8\ \mu s)^{-1}$ and $\kappa_\varphi = (313 \pm 40\ \mu s)^{-1}$. Our

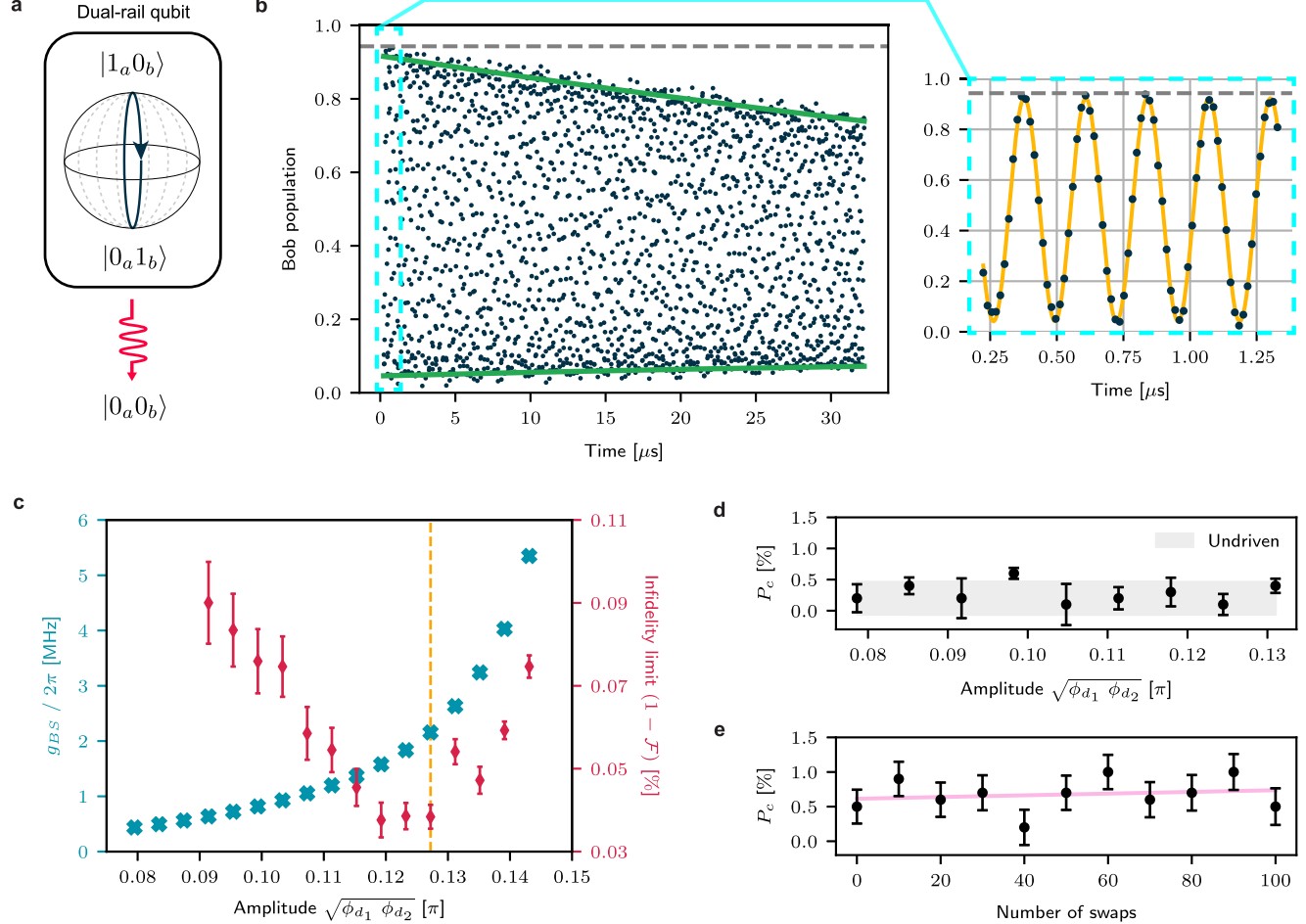

**Fig. 2 | Beamsplitting with the differentially-driven SQUID. a** Beamsplitting implements an effective driven Rabi evolution in the Bloch sphere of the dual-rail qubit formed by the single photon subspace Alice and Bob, where decay can be detected by monitoring the vacuum state. **b** Resonant evolution of a single-photon prepared in Bob. The data is normalized for readout infidelity, and state preparation fidelity is shown as a dashed gray line (Supplementary Note 10). The fast coherent oscillations (black dots) between the cavities are fitted to Eq. (6) (green lines show envelope) to obtain the decay and dephasing time-scales. The evolution for the first 1.5 μs is plotted separately to better illustrate the oscillations, and fit to a sinusoid to extract $g_{BS}$. **c** Sweeping both drive amplitudes simultaneously and repeating experiment (**a**) lets us quantify $g_{BS}$ (blue crosses), and the decoherence limit on beamsplitter infidelity (red diamonds) at various drive strengths. We

choose a drive strength with simultaneously low infidelity and high beamsplitter rate as our operating point (yellow dashed line). **d** The coupler's driven excitation ($P_c$) after evolving for 10 swaps is directly quantified through a dedicated on-chip readout mode. We observe no monotonic correlation with respect to drive amplitude, and driven populations mostly remain within the range of the undriven population (gray region). **e** Coupler population as a function of number of swaps at the operational driving point. The heating rate is nearly immeasurable, with a fitted (pink line) slope of $(1.2 \pm 2.4) \times 10^{-5}$ excitation per swap, which is within expectation for our natural thermal background ($\gamma_{c,\uparrow} - (3.3 \text{ ms})^{-1}$). The non-zero offset of the fit arises from preparation and readout infidelities. Error bars in both (**d**) and (**e**) represent fit errors from the protocol described in ref. 46.

effective $\kappa_{BS} = (150 \pm 25 \text{ μs})^{-1}$ places a decoherence-based upper bound on the fidelity of $\mathcal{F} = 99.96 \pm 0.01\%$, which is almost two orders of magnitude better than the previous transmon-based implementation. Crucially, this fidelity is also limited primarily by photon loss in the cavities, preserving their advantageous noise bias.

To directly quantify the suppression of drive-induced coupler excitation, we measure the coupler population as a function of drive amplitude after resonantly evolving for ten swaps. With our coupler prepared in the ground state, we apply this pulse and measure the coupler's population through a protocol that is robust to readout infidelity[46]. We measure no correlated increase of its driven population as a function of drive amplitude, up to our measurement uncertainty of ~0.2% (Fig. 2d). At the operating point, we explicitly quantify the increase in coupler excitation as a function of number of swaps (Fig. 2e). We evolve the system up to 100 swaps and observe a total heating rate below ~$4 \times 10^{-5}$ excitations per swap, which is consistent with the undriven heating rate of the coupler, implying no additional

drive-induced heating. This substantial suppression (three orders of magnitude better than transmon-based implementations[26]) eliminates limitations placed by coupler-induced dephasing on the fidelity of the beamsplitter, allowing us to harness the long lifetimes and even longer dephasing times of the 3D cavities.

While this device was optimized for performance in the single-photon manifold, any drive-induced increase in the self-Kerr of the cavities would lead to coherent errors in higher-photon manifolds. The sideband interaction resulting from our choice of mode frequencies exacerbates this effect, with up to 128 KHz of inherited Kerr at the operating point. Numerous avenues exist to minimize this driven nonlinearity if desired (see Supplementary Note 6), including choosing a slightly higher coupler frequency, arraying multiple SQUIDs, or dynamical Kerr cancellation[47]. If minimizing inherited self-Kerr is vital, such as in schemes utilizing large coherent states, then implementing these improvements or using an alternative scheme like Kerr-free three-wave mixing[22,48] may be required.

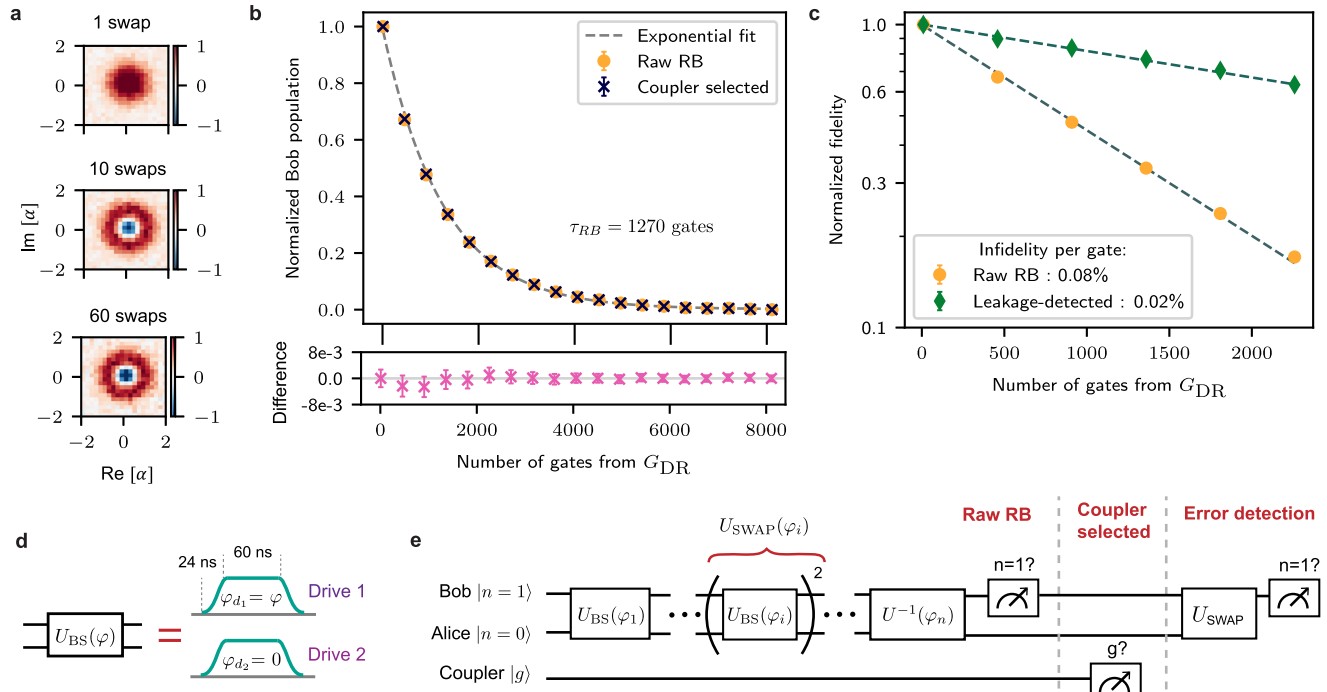

**Fig. 3 | Randomized benchmarking with a calibrated beamsplitter pulse.**
**a** Wigner function of Bob after preparing $|0_a1_b\rangle$ and implementing 1, 10 and 60 calibrated swaps. Each swap is a combination of two identical beamsplitter pulses. **b** Probability of ending in the target state $|0_a1_b\rangle$ after executing the RB protocol with randomly selected gates from $G_{DR}$ (yellow circles). The curve is normalized to account for state preparation and readout imperfections (Supplementary Note 10), and is in good agreement with a single-exponential, with a decay constant of $\tau_{RB} = 1271 \pm 4$ gates. This 'Raw RB' is practically indistinguishable from the sequences post-selected on the coupler ground state (black crosses), with the difference of the two curves shown below the main plot (pink crosses). **c** Focusing on the first 2250 gates, we use measurements of both cavities to post-select on sequences in which no photon loss event occurred (green diamonds). We compare

these sequences to the raw RB in (**b**) (yellow), showing an improvement in average gate infidelity from $0.078 \pm 0.001\%$ to $0.020 \pm 0.001\%$. **d** The gate-sets required for the above protocols are generated from calibrated beamsplitter pulses with tanh-shaped ramps, where different $U_{BS}(\varphi)$ are obtained by changing the relative phase of our drives. **e** The benchmarking sequences consist of randomly generated pulses that, under ideal operation, map $|0_a1_b\rangle$ back to itself. After each sequence, we measure whether the coupler is in its ground state, the presence of a photon in Bob, and the presence of a photon in Alice using an additional swap gate. This lets us generate the raw, coupler-selected and leakage-detected RB datasets. All sequences are also conditioned on Bob's ancilla ending in its ground state, to discount first-order effects of ancilla heating.

## Benchmarking fidelity in the single-photon subspace

We precisely characterize the fidelity and noise bias of our beamsplitter by using it to implement universal control of the dual-rail qubit subspace, allowing techniques akin to standard randomized benchmarking (RB) protocols[49–51]. First, we identify the amplitude and frequency required for a fixed-length pulse (Fig. 3d) to achieve the beamsplitter unitary

$$U_{BS}(\varphi = 0) = e^{i\pi/4(\hat{a}^\dagger\hat{b} + \hat{a}\hat{b}^\dagger)}, \tag{8}$$

by repeating the pulse to perform swaps between Alice and Bob, and iteratively checking its performance up to ~1000 such repetitions (Supplementary Note 8). This calibrates an effective $X_{\pi/2}$ gate for the dual-rail qubit, with a relative amplitude precision of less than $3 \times 10^{-6}$ and frequency precision of less than $3 \times 10^{-7}$. Through phase control and repetition, we use this pulse to construct a set of native gates,

$$G_{DR} = \{X_{\pi/2}, Y_{\pi/2}, X_{-\pi/2}, Y_{-\pi/2}, X_\pi, Y_\pi\}, \tag{9}$$

that generate the Clifford group for the dual-rail qubit (Fig. 3d, Supplementary Note 9). This allows a form of direct randomized benchmarking[52], which under uniform sampling should convert both dephasing and coherent control errors into an effective depolarization channel. The dominant but detectable error of cavity photon loss appears as a leakage to the orthogonal state $|0_a0_b\rangle$, which is not converted to depolarization under this protocol, but can be separately quantified and selected out in post-processing.

The RB protocol consists of initializing the system in $|0_a1_b\rangle$ with the coupler and ancilla prepared in their ground states, and running sequences of varying lengths of randomly chosen gates from $G_{DR}$. Each sequence ends with an additional gate from $G_{DR}$ that maps the state back to $|0_a1_b\rangle$, after which the presence of an excitation in the coupler, Bob, and Alice are measured (Fig. 3e). We explicitly discount the effects of the ancilla in all results shown by separately measuring the ancilla and only including sequences where it ends in the ground state. The sequences range up to 8100 gates, where we choose up to 900 random gates (limited by FPGA memory), and repeat each gate nine times to fully capture the fidelity decay timescale (this is roughly equivalent to performing nine times as many random gates and provides a lower-bound for the fidelity of a non-repeated sequence, see Supplementary Note 9). We average over ~$10^5$ such semi-random sequences at each sequence length.

We observe that the average success probability of returning to Bob (Fig. 3b) decays exponentially, with an effective time constant of $\tau_{RB} = 1271 \pm 4$ gates. This curve is normalized to account for effects from state preparation and readout infidelity. We compare this curve to post-selected sequences where the coupler explicitly ended in the ground state, finding remarkable similarity. This comparison strongly suggests that the coupler-heating induced errors on actual beamsplitter-based gates have been suppressed to below the measurement accuracy, agreeing with our earlier measurement of diminished driven coupler excitation.

Finally, we quantify our beamsplitter infidelity and noise bias by comparing the raw protocol to sequences where the system was

observed to have not experienced a photon loss error. We focus on the span of the first 2250 gates for this comparison (Fig. 3c, plotted on a log scale), where the raw decay of Bob's population suggests an un-selected single gate infidelity of 0.078 ± 0.001%. On detecting out leakage events, our evolution changes to a curve that decays toward a steady-state fidelity of -0.5 instead of 0, as expected (Supplementary Figure 7). For a fair comparison between this curve and the raw RB, we re-scale this fidelity to also have a steady-state of zero (Supplementary Note 10). This helps illustrate the clear improvement under error-detection to an infidelity per gate of 0.020 ± 0.001%, which shows that discarding only one out of every ~1300 shots per gate can lead to a 3.9 ± 0.2 × increase in gate performance, enabled by the cavity noise-bias. We reiterate that the fact that the dominant errors remain photon loss, which is detectable or correctable with various bosonic encoding schemes, satisfies one of the key goals for a high-performance inter-cavity interaction.

Notably, because our gate-set is crafted from nearly identical beamsplitter pulses, with a random gate from $G_{DR}$ containing 4/3 beamsplitters on average, we can directly convert these gate infidelities into an effective single beamsplitter fidelity. Our measurements thus imply an effective un-selected beamsplitter fidelity of 99.941 ± 0.001%, which improves on leakage detection to 99.985 ± 0.001%. The remaining errors after leakage detection can be due to intrinsic dephasing of the cavities, drifts in our control electronics, or other effects that are not treated by the post-selection protocols, like cascaded heating and decay events of the ancilla[12].

## Discussion

In conclusion, we constructed a microwave implementation of a tunable cavity-cavity beamsplitter and characterized its performance within the cavities' joint single-photon subspace. We obtained a beamsplitter gate fidelity exceeding 99.94% in this subspace, and were limited by detectable single-photon loss in the cavities. This performance was enabled by carefully engineering the drive frequencies and leveraging fundamental symmetries of the nonlinear converter to keep the coupling mode in its ground state even when driving a fast beamsplitter. Such high-fidelity control of a strongly driven nonlinear element in the presence of decoherence is a significant step forward for fast parametric operations in circuit-QED.

While this system was optimized to engineer a beamsplitter between 3D cavities, the generality and versatility of the design framework far exceed this specific application. One could apply such a beamsplitter to on-chip resonators, or to phononic modes in hybrid architectures. Other low-order mixing processes allowed by the converter's symmetry, such as two-mode or single-mode squeezing, could also be implemented while still reaping the benefits of parity protection and suppressed coupler-induced infidelity. Parity protection is also not exclusive to the SQUID Hamiltonian[21,32], and future converter designs could leverage more advantageous forms of this selection rule.

Beyond the context of parametric interactions, we have also demonstrated the delivery of AC flux in a high-Q 3D environment. This can be used to control other devices in similar architectures that require fast-flux modulation. The electromagnetic simulation techniques we have developed are also readily applicable to other work involving driven circuit-QED systems, for both 3D and planar devices.

Finally, the demonstration of high-fidelity control in the dual-rail subspace motivates the hypothesis that this subspace could itself be used as a computational qubit[12,24,36]. The error-hierarchy of detectable decay over dephasing makes the dual-rail qubit amenable to erasure conversion[53], an approach potentially yielding high thresholds in the surface-code architecture. The single-qubit control demonstrated here can be extended to realize a high-fidelity gate-set[17] for multi-qubit control, charting a path towards a general dual-rail qubit-based architecture in circuit-QED. The performance in higher-photon manifolds is the subject of further research, with promising avenues

including arraying the SQUID element, and creating parity-protected couplers with suppressed Kerr nonlinearity.

## Methods

### Deriving the programmable beamsplitter Hamiltonian

We now show how differential driving at the correct resonance condition generates a beamsplitter between Alice and Bob. For the sake of simplicity in derivations, we set $\hbar = 1$. The differentially-driven SQUID Hamiltonian in Eq. (1) can be expanded in the ladder operators of the bare SQUID coupler ($\hat{\tilde{c}}, \hat{\tilde{c}}^\dagger$), in a frame rotating at the coupler frequency ($\omega_c$), and gives (up to quartic order):

$$\hat{\mathcal{H}}_{tot}^{(4)} = -\frac{E_C}{2}\hat{\tilde{c}}^{\dagger 2}\hat{\tilde{c}}^2$$
$$+ E_J(\cos(\phi_d) - 1)\theta_{c,zpf}^2\hat{\tilde{c}}^\dagger\hat{\tilde{c}}. \tag{10}$$

Here, $\phi_d$ is assumed to be driven at frequencies far off-resonant from the coupler squeezing condition, and $\theta_{c,zpf} = \left(\frac{2E_C}{E_J}\right)^{\frac{1}{4}}$ is the zero-point fluctuations of the bare coupler phase.

The drive ($\phi_d$) modulates the bare coupler frequency, which enacts a beamsplitting process between the cavities Alice and Bob ($\omega_{a,b}$) through their linear capacitive couplings to the coupler ($\omega_c$):

$$\hat{\mathcal{H}}_{\text{coupling}} = \sum_{\hat{\tilde{k}}=\hat{\tilde{a}},\hat{\tilde{b}}} g_k(\hat{\tilde{k}} + \hat{\tilde{k}}^\dagger)(\hat{\tilde{c}} + \hat{\tilde{c}}^\dagger)$$

Here $\hat{\tilde{k}} = \hat{\tilde{a}}, \hat{\tilde{b}}$ represent the bare modes of Alice and Bob, respectively. The coupling strengths $g_{a,b}/2\pi \approx 80$ MHz are designed to place us in the dispersive regime ($\Delta_{a,b} = \omega_{a,b} - \omega_c \gg g_{a,b}, E_C$) and therefore we can follow standard circuit-QED derivations[54,55] to diagonalize the linear part of the Hamiltonian, yielding Eq. (2) in the main text. We then move into a frame rotating at the dressed frequencies of Alice, Bob, and the coupler, respectively, and keep relevant terms:

$$\hat{\mathcal{H}}_{tot}^{(RWA)} = \hat{\mathcal{H}}_{static}^{(RWA)} + \hat{\mathcal{H}}_{driven}^{(RWA)}$$
$$\hat{\mathcal{H}}_{static}^{(RWA)} = -\sum_{k=a,b} 2E_C\beta_k^2\hat{c}^\dagger\hat{c}\hat{k}^\dagger\hat{k} - \frac{E_C}{2}\hat{c}^{\dagger 2}\hat{c}^2$$
$$\hat{\mathcal{H}}_{driven}^{(RWA)} = \frac{\omega_c}{2}(\cos(\phi_d) - 1)$$
$$\times \sum_{k,k'=a,b,c} \beta_k\beta_{k'}\hat{k}^\dagger\hat{k}'e^{i(\omega_k-\omega_{k'})t} \tag{11}$$

where $\hat{a}, \hat{b}, \hat{c}$ are the ladder operators for the dressed Alice, Bob, and coupler modes, respectively, $\beta_{a,b} = \frac{g_{a,b}}{\Delta_{a,b}}$ are the participations of Alice and Bob in the coupler mode, and $\beta_c \approx 1$.

Next, we introduce a pair of drive tones whose response on the actuator is given by:

$$\phi_d = \sum_{1,2}\phi_{d_{1,2}}(t) = \sum_{1,2}|\phi_{d_{1,2}}|\sin\left(\omega_{d_{1,2}}t + \varphi_{d_{1,2}}\right) \tag{12}$$

which are IQ-modulated signals, with Local Oscillators (LOs) that are directly derived by down-converting Bob and Alice's LOs, respectively, using a reference signal $\omega_\Delta$ (see Supplementary Note 13). This places the difference of the two drive frequencies close to the Alice-Bob detuning by design:

$$\omega_{d_{2,1}} = \omega_\Delta - \omega_{a,b} + \Delta_{d_{2,1}}^{(ssb)} \tag{13}$$

$$\Rightarrow \Delta_d = \omega_{d_2} - \omega_{d_1} = \Delta_{ab} + \Delta_{d_2}^{(ssb)} - \Delta_{d_1}^{(ssb)} \tag{14}$$

Here $\omega_\Delta \approx \omega_a + \omega_b - \omega_{\text{buffer}}$ is chosen to place the drive tones symmetrically about the buffer mode resonance, and $\Delta_{d_{1,2}}^{(\text{ssb})}$ are the detunings in the sideband modulation frequency for each drive tone with respect to the sidebands used to control Alice and Bob, respectively. We set $\Delta_{d_2}^{(\text{ssb})} = 0$ and use $\Delta_{d_1}^{(\text{ssb})} \ll \Delta_d$ as a finely tunable parameter to find the true beamsplitter resonance condition. Similarly, the drive phases ($\varphi_{d_{1,2}}$) are defined with respect to the control of Alice and Bob, respectively, with $\varphi_{d_2} = 0$ and $\varphi_{d_1} = \{0, \pi/2, \pi, -\pi/2\}$ being used to generate the various pulses required for the randomized benchmarking sequence.

In this limit of $\Delta_d \approx \Delta_{ab} \ll \omega_{d_1} + \omega_{d_2}$, we can simplify the drive using the Jacobi-Anger expansion and only keep terms that are either static or rotating at the detuning frequency (up to second order in drive strength):

$$\cos(\phi_d) \approx J_0\left(|\phi_{d_1}|\right) J_0\left(|\phi_{d_2}|\right) \\ + 2 J_1\left(|\phi_{d_1}|\right) J_1\left(|\phi_{d_2}|\right) \cos\left(\Delta_d t + \varphi_{d_1}\right), \tag{15}$$

where $J_0$ and $J_1$ are the zeroth and first-order Bessel functions. Substituting this into Eq. (11), we find, keeping only slow-rotating terms:

$$\hat{\mathcal{H}}_{\text{driven}}^{(\text{RWA})} \approx \frac{\omega_c}{2}\left[J_0\left(|\phi_{d_1}|\right) J_0\left(|\phi_{d_2}|\right) - 1\right] \sum_{k=a,b,c} \beta_k^2 \hat{k}^\dagger \hat{k} \\ + \frac{\omega_c}{2} J_1\left(|\phi_{d_1}|\right) J_1\left(|\phi_{d_2}|\right) \beta_a \beta_b \\ \times \left(e^{i(\Delta_d - \Delta_{ab})t + \varphi_{d1}} \hat{a}\hat{b}^\dagger + \text{h.c.}\right) \tag{16}$$

Importantly, this approximation only holds when the drive tones don't activate any other higher-order resonances. The first part of this driven Hamiltonian is a driven frequency shift, which we call the AC-Zeeman shift, and is reminiscent of the Stark shift in a regular charge-driven transmon. This is easier to see when rewriting it for small amplitude, where $(1 - J_0(x)) \approx x^2/4$:

$$\Delta_{Z,k} = -\beta_k^2 E_C\left(\frac{|\phi_{d_1}|^2 + |\phi_{d_2}|^2}{\theta_{c,\text{zpf}}^2}\right) \tag{17}$$

Specifically, the frequency shift of the coupler ($\Delta_{Z,c}$) provides a simple way to experimentally calibrate the strength of the drives (see Fig. 4a).

Finally, we derive the beamsplitting Hamiltonian by assuming the coupler to be in the ground state ($\langle c^\dagger c\rangle = 0$), which eliminates contributions from $\hat{\mathcal{H}}_{\text{static}}^{(\text{RWA})}$. We move into frames rotating at Zeeman-shifted frequencies for both Alice and Bob ($\omega_{a,b} + \Delta_{Z,(a,b)}$). We carry out an additional frame transformation for Alice, to a frame rotating at $\Delta_{\text{BS}} = \Delta_{ab} - \Delta_d + \Delta_{Z,ab}$, which gives us the beamsplitter Hamiltonian:

$$\hat{\mathcal{H}}_{\text{BS}}/\hbar = \Delta_{\text{BS}} \hat{a}^\dagger \hat{a} + g_{\text{BS}}\left(e^{i\varphi_{\text{BS}}} \hat{a}^\dagger \hat{b} + \text{h.c.}\right). \tag{18}$$

Here, $\varphi_{\text{BS}} = \varphi_{d_1}$, $\Delta_{Z,ab} = \Delta_{Z,a} - \Delta_{Z,b}$ is the relative Zeeman shift of the cavities, and $g_{\text{BS}}$ is the value described in the main text in Eq. (4).

We find $\Delta_{Z,ab}$ to be on the order of $g_{\text{BS}}$, which needs to be taken into account when finding the resonance condition. We approximately calibrate for it by preparing a single photon in Bob and examining the system's resonant evolution as a function of $\omega_{d_1}$. This produces a Chevron-like pattern (Fig. 4b), whose oscillations follow a detuned Rabi model: $\omega_{\text{osc}} = \sqrt{4g_{\text{BS}}^2 + \Delta_{\text{BS}}^2}$. Fitting to this model (Fig. 4c) lets us find both the $g_{\text{BS}}$ at any amplitude and the drive detuning required for $\Delta_{\text{BS}} = 0$.

## Fine-tuning a differential drive

We realize a purely differential drive by utilizing the separation of the coupler and actuator in electromagnetic space—the coupler has a large electric dipole moment, while the actuator appears primarily as a magnetic dipole. This provides a convenient way to address these two modes in a 3D architecture, which is not immediately present in other parity-protected devices like the asymmetrically threaded SQUID (ATS)[32] or the Josephson parametric converter (JPC)[21]. We require our drive to imitate a spatially uniform AC flux in the SQUID loop, with minimal stray capacitive coupling to its electric dipole. The introduction of such a high-frequency flux-bias line into a superconducting package is challenging with existing techniques, since it must both preserve the lifetime of the cavities and also effectively screen parasitic driven electric fields.

We are able to conveniently address both of these issues by utilizing the natural geometric separation of the electric and magnetic fields in a $\lambda/4$ co-axial post-cavity. We engineer our drive antenna as a 3D cavity that we call the 'buffer mode', that is incorporated into the same monolithic package that contains our high-Q storage cavities (Fig. 1b), thus introducing minimal additional seam or dielectric loss.

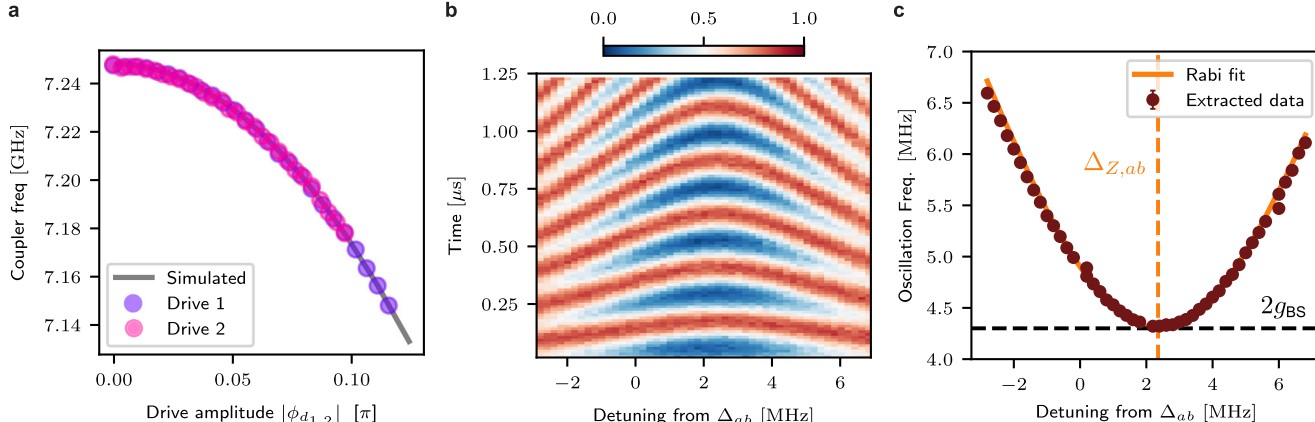

**Fig. 4 | Measured driven Zeeman shift and cavity swaps. a** We measure the coupler frequency through direct spectroscopy under a single drive tone, as a function drive amplitude. Comparing the Zeeman shift for either drive tone (pink, purple) to the prediction from Floquet simulation (gray solid line) allows accurate calibration of the drive strength in terms of the driven junction phase $\phi_{d1,2}$. **b** Measured population in Bob in the presence of both drive tones as a function of the drive-detuning ($\Delta_d - \Delta_{ab}$) and the time of evolution. A single photon is prepared in Bob, and the drives swap this photon between Alice and Bob under the detuned beamsplitter interaction. **c** Fitting the oscillations as a function of drive-detuning to the Rabi model allows us to calibrate amplitude-dependent shift in the resonance condition ($\Delta_{Z,ab}$) and strength $g_{\text{BS}}$ of the beamsplitting interaction.

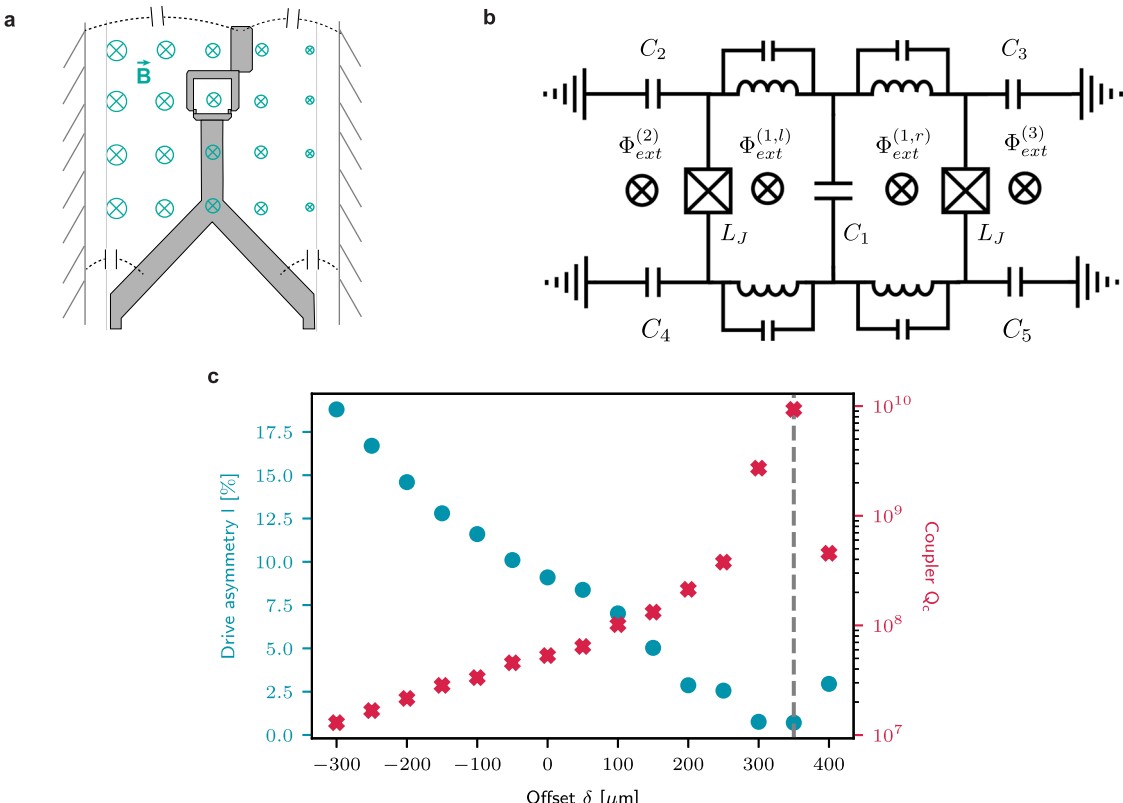

**Fig. 5 | Minimizing residual drive asymmetry in the SQUID via fine-tuning the circuit geometry. a** The SQUID device in the buffer cavity package driven by an oscillating B-field. The cylindrical geometry of the buffer cavity (outer wall radius greater than the inner wall) dictates that the B-field has a non-uniform distribution along the radial direction (from left to right). The dashed lines with capacitor symbol represents the capacitance between the antenna pads of the SQUID and the wall of the package. **b** The lumped-element circuit model of the SQUID device, taking into account the geometry of SQUID and the spatial distribution of the B field. **c** HFSS simulation of the quality factor of the common mode (red) and drive asymmetry (blue, defined in Eq. (21)) as functions of the top pad displacement from center to right, $\delta$. As a result of introducing this asymmetry in the device geometry, the optimization of the drive asymmetry and the quality factor are simultaneously achieved at $\delta \approx 350\ \mu$m.

The base of this buffer cavity forms a virtual electric ground where we simultaneously have an E-field node and a B-field anti-node. Placing the SQUID loop in this region, with its electric dipole moment oriented perpendicular to the direction of any residual E-field, automatically smothers any stray capacitive coupling between the buffer mode and the coupler.

Any additional effects due to a spatial non-uniformity of the magnetic field threading the loop can be addressed through a purposely-engineered asymmetric capacitance in the on-chip SQUID device (Fig. 5a). This asymmetry in the capacitance serves as a control knob for tuning the coupling between the electromotive force and the coupler mode in the presence of an alternating magnetic field, allowing to compensate for residual drive asymmetry[56]. To illustrate this effect, we analyze the circuit model in Fig. 5b that captures this effect to represent the actual experimental device. We take into account a non-uniform AC magnetic field that is distributed across the device, not only inducing a flux in the SQUID loop, but also inducing an electromotive force on the shunting capacitors. Assuming the geometric inductance of the loop is much smaller than the Josephson inductance (Supplementary Note 1), we arrive at the effective Hamiltonian of

$$\hat{\mathcal{H}} = \frac{\left(\hat{Q}_c - C_\Sigma V_{\mathrm{emf}}\right)^2}{2C_\Sigma} - E_J \cos\frac{\Phi_{\mathrm{ext}}^{(1,\Sigma)}}{2\phi_0}\cos\frac{\hat{\Phi}_c}{\phi_0}. \quad (19)$$

Here, $\hat{Q}_c$ is the charge operator of the common mode, $C_\Sigma = C_1 + C_2 + C_3 + C_4 + C_5$ is its total shunting capacitance, and

$\Phi_{\mathrm{ext}}^{(1,\Sigma)} = \Phi_{\mathrm{ext}}^{(1,l)} + \Phi_{\mathrm{ext}}^{(1,r)}$ is the total external flux penetrating the SQUID loop. The charge operator $\hat{Q}_c$ and the flux operator $\hat{\Phi}_c$ are related to the Cooper-pair number operator $\hat{n}_c$ and the superconducting phase operator $\hat{\theta}_c$ through $\hat{Q}_c = 2e\hat{n}_c$, $\hat{\Phi}_c = \phi_0\hat{\theta}_c$, where $e$ and $\phi_0$ are the electron charge and the reduced flux quantum, respectively. The electromotive voltage, $V_{\mathrm{emf}}$, is given by

$$V_{\mathrm{emf}} = \frac{C_1}{2C_\Sigma}\dot{\Phi}_{\mathrm{ext}}^{(1,\delta)} + \frac{C_l}{C_\Sigma}\left(\frac{1}{2}\dot{\Phi}_{\mathrm{ext}}^{(1,\Sigma)} + \dot{\Phi}_{\mathrm{ext}}^{(2)}\right) - \frac{C_r}{C_\Sigma}\left(\frac{1}{2}\dot{\Phi}_{\mathrm{ext}}^{(1,\Sigma)} + \dot{\Phi}_{\mathrm{ext}}^{(3)}\right), \quad (20)$$

with $\Phi_{\mathrm{ext}}^{(1,\delta)} = \Phi_{\mathrm{ext}}^{(1,l)} - \Phi_{\mathrm{ext}}^{(1,r)}$, $C_l = C_2 + C_4$, $C_r = C_3 + C_5$.

Shifting the upper antenna pad of the SQUID from the center to the right simultaneously increases $C_3$ and decreases $C_1$ and $C_2$ in Eq. (20). Fine-tuning this offset ($\delta$) can result in the minimization of $V_{\mathrm{emf}}$, as demonstrated in Fig. 5c, where we simulate the drive asymmetry $l$ (Supplementary Note 2) at various offsets in Ansys finite-element High-Frequency Simulation Software (HFSS). We obtain this asymmetry from the currents across the two junctions,

$$l = \frac{|\phi_c|}{\sqrt{|\phi_c|^2 + |\phi_d|^2}} = \frac{|I_{J1} + I_{J2}|}{\sqrt{2\left[|I_{J1}|^2 + |I_{J2}|^2\right]}}, \quad (21)$$

with $I_{J1,2} = \phi_0\phi_{1,2}/L_{J1,2}$. When the pad is shifted towards the right by $\delta \approx 350\ \mu$m, the asymmetry in the drive is almost completely eliminated, resulting in the desired purely differential drive. At the same

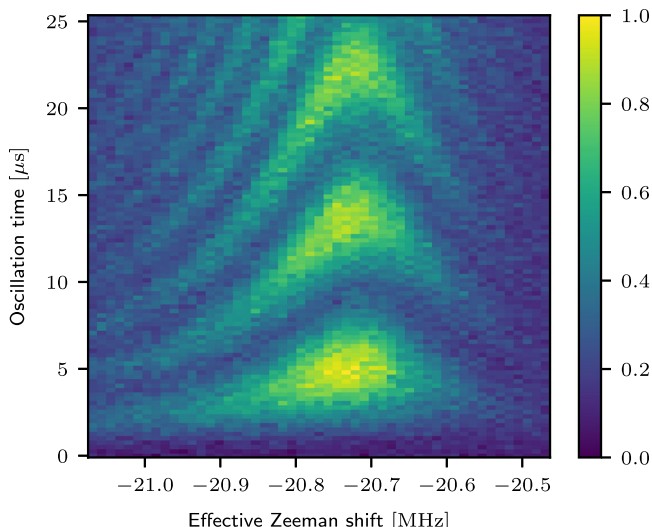

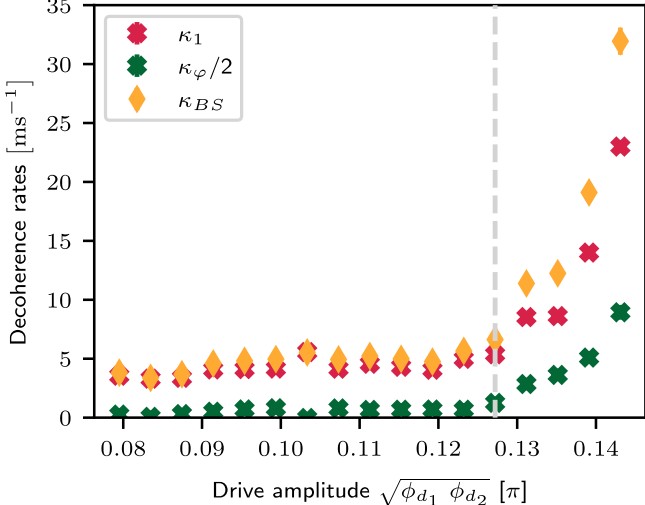

**Fig. 6 | Experimentally characterizing drive asymmetry.** We directly probe the coupler 'ge/3' transition due to any residual common-mode drive. The coupler is prepared in the ground state at $t = 0$ μs, followed by a single-tone drive through the buffer-mode near $\omega_c/3$. The rate of coherent oscillation of the coupler population (color-scale) compared to the effective Zeeman shift (x-axis) of this resonance frequency bounds the drive asymmetry to <1%.

**Fig. 7 | Measured driven decoherence.** Decoherence rates are extracted from short sections of the long-time evolution under both drive tones, as a function of drive amplitude. The sideband collision of coupler and Bob (see Supplementary Note 5) clearly limits the fidelity at high amplitudes, and we operate on the boundary of this collision (gray dashed line), where $\kappa_{BS}$ is limited by $\kappa_1$.

time, the quality factor of the common mode (coupler) reaches a maximum, since the Purcell effect from the coupler's linear coupling to the buffer mode is highly suppressed. In the experiment, this lets us strongly couple the buffer mode to a coaxial drive line and achieve a quality factor (Q) as low as 12 while maintaining a $Q_c \sim 2 \times 10^6$ for the coupler, and $Q_a \sim 1.4 \times 10^7$, $Q_b \sim 1.2 \times 10^7$ for Alice and Bob, respectively. We also make sure that other types of back-actions of the buffer mode, such as the coherent- or thermal-photon-induced dephasing, is not limiting the coherence of the coupler and the high-Q storage modes (Supplementary Note 3).

### Experimentally characterizing residual drive asymmetry

Non-idealities in the experimental implementation of our differential drive inevitably result in a small but finite common-mode drive strength. One way to experimentally characterize this residual drive asymmetry is to probe the strength of interactions that are uniquely generated by the common-mode drive. A candidate for this is the 'ge/3' interaction, where a single drive tone at $\approx \omega_c/3$ drives Rabi oscillations of the coupler between its ground state and its excited state through three drive photons. The rate of this Rabi-like process is given by (Supplementary Note 1):

$$\Omega_{\text{ge}/3} = \frac{E_J}{h} \theta_{c,\text{zpf}} \phi_c \left( |\phi_c|^2/3 + |\phi_d|^2 \right). \tag{22}$$

To obtain the relative drive asymmetry, we compare the strength of the ge/3 process to the Zeeman shift induced by the same single-tone drive, which depends on both $\phi_c$ and $\phi_d$:

$$\Delta_{Z,c} \approx \left[ J_0(|\phi_c|) J_0(|\phi_d|) - 1 \right] \omega_c/2 \tag{23}$$
$$\approx -\frac{\omega_c}{8} \left( |\phi_c|^2 + |\phi_d|^2 \right).$$

Note the difference between this and Eq. (17) which assumes a purely differential drive with two tones. Measuring the strength of both these processes (Eqs. (22) and (23)) is sufficient to extract $|\phi_c|$ and $|\phi_d|$. As illustrated in Fig. 6, we observe the ge/3 Rabi oscillation of the coupler with a rate of $\Omega_{\text{ge}/3}/2\pi = 0.12$ MHz. The resonant frequency of the ge/3

oscillation is at 2.408 GHz, which corresponds to an effective Zeeman shift of $\Delta_{Z,c}/2\pi = -21$ MHz. Using Eqs. (22) and (23), we obtain $|\frac{\phi_c}{\phi_d}| \approx 2.5e - 3$, corresponding to a drive asymmetry of $l < 1\%$. It is possible that the drive asymmetry is a little higher in the actual experiment, due to a deviation from our assumption of perfectly symmetric junction energies, but we expect this deviation to be small (Supplementary Note 7).

### Decoherence in the dual-rail subspace

We can derive the expected evolution and the beamsplitter fidelity from the time evolution of the density matrix in the dual-rail subspace (Supplementary Note 4):

$$\rho_{\text{DR}}(t) = \frac{1}{2} e^{-\kappa_1 t} \times \begin{bmatrix} 1 + e^{-\kappa_\varphi t} \cos(\Omega t) & i \sin(\Omega t) e^{-\kappa_\varphi t} \\ -i \sin(\Omega t) e^{-\kappa_\varphi t} & 1 - e^{-\kappa_\varphi t} \cos(\Omega t) \end{bmatrix}, \tag{24}$$

where we have used $\Omega = 2g_{BS}$ for simplicity. The upper left element measures the probability of finding a photon in Bob, which is Eq. (6) in the main text.

To calculate the beamsplitter fidelity, we can set the evolution time to be $t_{BS} = \frac{\pi}{4g_{BS}}$ and calculate the overlap with a lossless evolution $(\rho^{(0)}_{|\psi^\pm\rangle})$ that has no leakage or dephasing:

$$\mathcal{F}(t_{BS}) = \text{Tr} \left[ \rho_{\text{DR}}(t_{BS}) \times \rho^{(0)}_{\text{DR}}(t_{BS}) \right]$$
$$= \frac{1}{2} \left[ e^{-\kappa_1 t} + e^{-(\kappa_1 + \kappa_\varphi)t} \right] \tag{25}$$
$$\approx 1 - \kappa_{BS} t_{BS}.$$

Here $\kappa_{BS} = \kappa_1 + \frac{\kappa_\varphi}{2}$, which gives us Eq. (7) in the main text.

In order to efficiently extract $\kappa_1$ and $\kappa_\phi$ from the experiment, we create a single photon in Bob, and measure Bob's population under the beamsplitter interaction within two sections of time: $t \in [0, \delta t]$, and $t \in [T, T + \delta t]$. For small $\delta t$ and large $T$, the measurement result within these time windows can be approximated as

$$P_0(t) = A \left( 1 + \cos(2g_{BS}t + \phi_0) \right) + B,$$
$$P_T(t) = A e^{-\kappa_1 T} \left( 1 + e^{-\kappa_\varphi T} \cos(2g_{BS}t + \phi_0) \right) + B. \tag{26}$$

Here, $A$ and $A + B$ represent the amplitude and the offset of the oscillation, which are close to but not strictly equal to 0.5 in the presence of state preparation and measurement (SPAM) errors. From fitting to $P_0(t)$ we extract $A$ and $B$, which are then used in the fitting of $P_T(t)$ to find $\kappa_1$ and $\kappa_\varphi$. We choose $T = 25\,\mu s$ and $\delta t \approx 20 t_{BS}$ to extract the decay rates at different drive amplitudes, shown in Fig. 7.

## Reporting summary

Further information on research design is available in the Nature Portfolio Reporting Summary linked to this article.

## Data availability

The data presented in this study is available at https://doi.org/10.6084/m9.figshare.23589144.v1 and more detailed source data is available from the corresponding authors upon request.

## Code availability

All computer codes used in this study are available from the corresponding authors upon request.

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

## Acknowledgements

We thank J. Venkatraman and X. Xiao for helpful conversations on parasitic nonlinear resonances. We are grateful to J. Curtis, A. Koottandavida, and I. Tsioutsios for technical assistance and providing useful code. We thank B. Chapman for providing parametric amplifiers used in this experiment, and A. Read for help with DC-flux line wiring. We are grateful to C. Wehr and H. M. Moseley for help with designing substrate clamps and K. Chou for advice on beamsplitter control. We thank A. Miano for help simulating the dc-flux spectroscopy of the SQUID. We also thank P. Winkel, M. Devoret, B. Chapman, S. de Graaf, S. Xue, J. D. Teoh, T. Tsunoda, S. Chakram, and J. Huang for useful discussions. This research was sponsored by the Army Research Office (ARO) under grant nos. W911NF-16-1-0349, W911NF-18-1-0212 and W911NF-23-1-0051. The views and conclusions contained in this document are those of the authors and should not be interpreted as representing the official policies, either expressed or implied, of the Army Research Office (ARO) or the US Government. Fabrication facilities use was supported by the Yale Institute for Nanoscience and Quantum Engineering (YINQE) and the Yale SEAS Cleanroom.

## Author contributions

Y.L. and A.M. designed and conducted the experiment, under the supervision of R.J.S. Y.L., A.M., and S.G. designed the experimental device and developed the differential drive scheme. Y.L. fabricated the DC-SQUID and ancilla transmon chip devices with help from S.G. and L.F. A.M., Y.L., and J.G. conducted measurements and analyzed the data. J.G., A.M., and Y.L. implemented the dual-rail qubit randomized benchmarking. Y.Z. and J.C. provided theoretical support for analyzing parametric processes and randomized benchmarking, respectively. S.M.G. provided important theoretical insights during the writing of the manuscript. A.M., Y.L., J.G., and R.J.S. wrote the manuscript with feedback from all co-authors.

## Competing interests

L.F. and R.J.S. are founders and shareholders of Quantum Circuits, Inc (QCI). S.M.G. receives consulting fees and is an equity holder in QCI. The remaining authors declare no competing interests.
