## [Peer Review File · Nature Communications]

High-fidelity parametric beamsplitting with a parity-protected converterREVIEWER COMMENTS

Reviewer #1 (Remarks to the Author):

I have now completed my review of the manuscript from Lu et al.. In this work, the authors propose, engineer, and demonstrate operation of a novel superconducting setup to perform beamsplitter operations between microwave photonic qubits encoded in a dual-rail basis. The beamsplitter operation is an integral component of the single-qubit gate set for dual-rail qubits, and the implementation developed by in this manuscript is high fidelity, with performance comparable to single qubit gates in other state-of-the-art systems such as superconducting qubits. To my knowledge, this is the highest fidelity beamsplitter demonstrated in microwave photonics, and this performance is due in large part to implementation of the differential drive, which suppresses many detrimental coherent processes. The careful and detailed engineering of the device as a whole further suppresses unwanted coherent and incoherent error processes.

The manuscript is technically sound, very well written, and exceptionally thorough. The authors have gone into great detail to explain the operation of their device, why it should have the beneficial properties they desire in theory, and all experimental results are justified with robust data analysis and statistics. One minor comment is that there is a lot of focus in the main text on spurious heating of the qubit and coupler modes, which does not appear to be the dominant error source of the demonstrated device. Upon reading the supplement it appeared to me that it might have been the case that heating was the dominant error source before further design refinement, but without this information a priori the focus on heating in the main text felt somewhat out of place.

This manuscript demonstrates an integral component in the microwave photonics toolbox, in a way that can certainly be replicated to produce multi-qubit devices. In my opinion, it will be the building block of much future research in this field. The demonstrated beamsplitter is high-fidelity, and the analysis and characterization are robust enough to support this claim. In light of its high impact within its domain, and likelihood of influencing future research directions I strongly recommend this manuscript for publication in Nature Communications.

I have a few technical comments regarding the benchmarking that I would recommend the authors address before publication. Note that I believe that the benchmarking results are sound enough for the claims made in the manuscript, but addressing the following comments would in my opinion improve the presentation of the benchmarking results.

1. I was confused by the choice of RB gate set, in particular the repetition of each beamsplitter gate nine times. The main text claims this was to “fully capture the leakage timescale”, which I can interpret in one of two ways. Either this is simply a probabilistic enhancement to increase the statistical signal of the leakage, i.e. enhance the decay rate of the non-post-selected data, or it is a pulse build-up effect where leakage is more likely for repeated gates than simple probabilistic arguments would imply. If the latter is the case, would this not also impact longer sequences (since all gates have the same underlying physical implementation) and cause (Clifford) gate-dependent error?

2. Are the fidelities quoted of the 9-beamsplitter cycle gates or of an individual beamsplitter gate? The manuscript reads to me like the former, since only the $4/3$ cycle gates per Clifford is discussed, but comparing the quoted error with the coherence limit the numbers seem more in line with an individual beamsplitter gate. The authors should clarify this in the manuscript, and if the latter is indeed what they are quoting, it should be noted that given that the error within a cycle is not twirled, extracting the error of an individual gate from the cycle error could have significant systematic bias if the underlying error is coherent. This would likely impact the leakage post-selected results more significantly.

3. Why did the authors not perform direct RB with a gateset composed of single beamsplitter gates? Did the authors try and it resulted in inconsistent results or non-exponential decay (perhaps due to leakage accumulation between gates)? I believe the manuscript would be improved by some further discussion of the reasoning for this non-standard benchmarking approach, perhaps in the methods or supplementary information.

4. One minor comment is that it would be useful to also plot error bars over or the sequence values of individual seeds, not just the mean of the RB decay curve. Showing the spread of individual sequences is useful for qualitatively assessing the coherence of the underlying

noise process.

Sincerely,

Luke Govia (signed intentionally in accordance with Nature's transparent peer review)

Reviewer #2 (Remarks to the Author):

Comments about “High-fidelity parametric beamsplitting with a parity-protected converter”

To the authors:

The manuscript by Lu *et al*/ describes the experimental demonstration of a high-fidelity gate between two 3D microwave cavities. The authors developed novel experimental techniques that allows them to exploit the natural symmetries of a dc-squid and use it as a parity protected coupler. The work is novel, and the discussion is clear, both on the theory and experiment side. Overall, the novelty/impact of the work and the quality of the manuscript meet the requirements for publication in Nature Communication.

Detailed comments:

- Line 36- 45: The deleterious effects of strong drive are also very much noticeable in amplifier type devices, so the statement seems a bit arbitrary. For example, “unwanted interaction with numerous [...] modes” is definitely an issue in other parametric systems, even low Q ones.
- Line 54-67: It is unclear at this point how the high Q resonator participate in θ_c . I would move here Eq 3 and associated text to help the discussion.
- Line 96: θ_1 and θ_2 are not defined.
- Line 139-161: This paragraph is kind of obscure without theoretical support. Also, not sure that a “squeezing resonance” is a thing. I would rephrase that this section.
- Line 293-297: why not GRAPE? Is there a trade-off in efficiency, robustness, simplicity?
- Line 322-323: why are you only fitting short sections and not the whole trace? Are there drifts?
- Line 329: giving an order of magnitude of the shift could be useful here (it is currently buried in SI)
- Line 345: the word ‘implies’ seems too strong. Here you use measured quantities to estimate an upper bound for the fidelity.
- Line 353-357: at this point it is unclear that the coupler has its own drive and readout line. Is the coupler prepared in g via a driven reset protocol as well? Would a drive and readout line be present on the coupler line in the future, or is it mainly here to investigate how well it works?
- Line 364: what are the units of the heating rate?
- Line 408: If I understand well, that means that now you post-select on the coupler being in the ground state, the ancilla being in the ground state, cavity photon loss, and number resolved measurement to prepare $|0_a 1_b\rangle$? Overall, what is the fraction of runs post-selected out?
- Line 432-449: the improvement under error-detection is useful if you post-select, but can it be error corrected?

- Section 4 in the SI: it would be useful to make a brief mention in the main text that the DC flux offset is calibrated out.
- Figure S5: Could you label what the different driving pins are for (ancilla/coupler drive/readout, etc)
- One overall comment is that the manuscript switches from theory to experimental/setup description a few times, making the manuscript a bit hard to follow. It would really help to finish the theory part before moving to the physical implementation.

Reviewer #3 (Remarks to the Author):

This article presents a high-fidelity beamsplitter coupling between two harmonic oscillator modes (3D EM-cavities), which can be turned on and off as it results from parametric driving. The process is achieved through the well-known Josephson junction nonlinearity, and a symmetric squid-architecture is used to select only even parity terms. The precision obtained on the 01-10 Fock space with this scheme is indeed very high.

I have two main points of criticism on the manuscript in its present form.

First, I somewhat fail to see the originality of the scheme compared to previous realizations like [15,33] cited by the authors. The main text remains rather vague, and the reference to `supp.mat.` is not focused enough. Therefore, it is not clear enough which main novel idea leads to this performance — or is it just the 3D cavities, plus restricting to 0,1 Fock space?

Second, I indeed find quite reductive to restrict the analysis to this 0,1 Fock space. While this may be one choice for qubit encoding, it is certainly not the only one, and it is certainly not the only inputs expected into a "beamsplitter". At higher Fock numbers, we may expect that nonlinearities and higher-order processes could significantly alter the functioning of the device, so probing 0,1 Fock is really not convincing evidence for claiming the announced beamsplitter Hamiltonian. In fact, for some readers it may even be quite suspicious to have no extensive comments on this.

Finally, as a minor comment, I would drop the post-selection part, or at least displace it to `supp.mat` only. Indeed, post-selection-based schemes always come with a grain of salt, so when this implies no too significant gain — like seems to be the case here — it is not helping your case to insist too much on this variant.

We have revised our manuscript with changes to respond to the reviewers' detailed comments. The two biggest changes in the manuscript were to separate much of the initial theoretical exposition from the introduction and move it to the supplement, and to better address the possible issues from driven nonlinearity in higher photon manifolds throughout the main text and in an additional supplemental section. All major changes have been summarized at the end of this reply.

We would like to take this opportunity to thank the reviewers for their detailed comments and for their positive remarks on the merit and quality of our manuscript. We believe that addressing their concerns significantly improved the quality of our manuscript, by both addressing key physics matters and making the main ideas and results more accessible. We now respond to their comments point-by-point.

Reviewer 1

I have now completed my review of the manuscript from Lu et al.. In this work, the authors propose, engineer, and demonstrate operation of a novel superconducting setup to perform beamsplitter operations between microwave photonic qubits encoded in a dual-rail basis. The beamsplitter operation is an integral component of the single-qubit gate set for dual-rail qubits, and the implementation developed by in this manuscript is high fidelity, with performance comparable to single qubit gates in other state-of-the-art systems such as superconducting qubits. To my knowledge, this is the highest fidelity beamsplitter demonstrated in microwave photonics, and this performance is due in large part to implementation of the differential drive, which suppresses many detrimental coherent processes. The careful and detailed engineering of the device as a whole further suppresses unwanted coherent and incoherent error processes.

The manuscript is technically sound, very well written, and exceptionally thorough. The authors have gone into great detail to explain the operation of their device, why it should have the beneficial properties they desire in theory, and all experimental results are justified with robust data analysis and statistics. One minor comment is that there is a lot of focus in the main text on spurious heating of the qubit and coupler modes, which does not appear to be the dominant error source of the demonstrated device. Upon reading the supplement it appeared to me that it might have been the case that heating was the dominant error source before further design refinement, but without this information a priori the focus on heating in the main text felt somewhat out of place.

This manuscript demonstrates an integral component in the microwave photonics toolbox, in a way that can certainly be replicated to produce multi-qubit devices. In my opinion, it will be the building block of much future research in this field. The demonstrated beamsplitter is high-fidelity, and the analysis and characterization are robust enough to support this claim. In light of its high impact within its domain, and likelihood of influencing future research directions I strongly recommend this manuscript for publication in Nature Communications.

I have a few technical comments regarding the benchmarking that I would recommend the authors address before publication. Note that I believe that the benchmarking results are sound enough for the claims made in the manuscript, but addressing the following comments would in my opinion improve the presentation of the benchmarking results.

We thank the reviewer for the appreciation of our work and the recommendation for publication in Nature Communications. The comments raised by the reviewer are addressed below.

1. ***“One minor comment is that there is a lot of focus in the main text on spurious heating of the qubit and coupler modes, which does not appear to be the dominant error source of the demonstrated device. Upon reading the supplement it appeared to me that it might have been the case that heating was the dominant error source before further design refinement, but without this information a priori the focus on heating in the main text felt somewhat out of place.”***

The spurious heating of the coupler mode when driven was indeed a primary roadblock in previous work [1], and its suppression in this work is what led to an improvement by ~50x in the achievable beamsplitter rate. In revising the introduction to better match *Nature Communications'* format we have aimed to make this connection more explicit in lines 54-58 of the main text.

[1] Y. Y. Gao *et al.* PRX 8, 021073 (2018)

2. ***“I was confused by the choice of RB gate set, in particular the repetition of each beamsplitter gate nine times. The main text claims this was to “fully capture the leakage timescale”, which I can interpret in one of two ways. Either this is simply a probabilistic enhancement to increase***

the statistical signal of the leakage, i.e. enhance the decay rate of the non-post-selected data, or it is a pulse build-up effect where leakage is more likely for repeated gates than simple probabilistic arguments would imply. If the latter is the case, would this not also impact longer sequences (since all gates have the same underlying physical implementation) and cause (Clifford) gate-dependent error?"

The motivation for this protocol is to increase the statistical signal of the fidelity decay, particularly the post-selected fidelity decay, simply by extending to longer sequence lengths. The impetus for specifically choosing to concatenate gates nine times is due to a rather technical constraint; our implementation of randomized benchmarking is limited by FPGA memory to a maximum sequence length of ~950 gates.

When fitting multi-parameter functions such as an exponential decay with an offset, we find it vital to have sufficient data to separate out uncertainties in the fit parameters. In order to properly fit the decay of an RB sequence, we need to capture several coherence times of the decay to accurately extract the offset due to imperfections in readout or residual thermal populations in the cavities. For a fidelity of ~99% (corresponding to a timescale of a few hundred gates) this limit would not have posed an issue. However, our exceptionally high fidelities warrant much longer sequence lengths on the order of ~1000 gates or more, requiring us to find a way past this FPGA limit.

In spite of this hardware constraint, a solution exists since this limit is specifically on the *number* of randomly selected operations, not the physical time of the total sequence. By re-defining one 'operation' to actually be $4N+1$ gates concatenated into one pulse we can extend our total sequence length out to several thousand gates and thus "fully capture the leakage timescale". We now realize the confusing wording of this phrase, and have changed it to "fully capture the fidelity decay timescale" as seen in lines 394-395.

We have updated our supplementary section on the Randomized Benchmarking protocol (now supplementary section 9) to include single-gate (no concatenation) RB data up to the FPGA limit overlaid with concatenated data in figure S7a. As shown in this figure, these single-gate curves would have provided insufficient information to properly fit both the decay rate and offset of the exponential, especially for the leakage-detected data. Additionally, we see that the concatenated pulses perform remarkably similarly to the single-gate pulses, so we don't believe that this concatenation causes any problematic gate-dependent error. For additional commentary on these curves and how they compare to their concatenated counterparts, please see our responses to further comments below.

For completeness, elaborating a bit further on the precise nature of the FPGA limit, our randomized benchmarking is implemented by first filling an array of registers containing wave addresses in memory with each address corresponding to a certain operation, and then stepping through this array to execute the sequence. The FPGA has memory for 1024 registers, and some

of these registers are dedicated to other functions (such as implementing a pseudo-random number generator), so there are only about 950 register locations left over for the array containing wave addresses. Updating our control electronics to use state-of-the-art FPGA hardware and thus far surpassing this memory limit (among many other improvements) is an active effort in our lab.

- 3. *“Are the fidelities quoted of the 9-beamsplitter cycle gates or of an individual beamsplitter gate? The manuscript reads to me like the former, since only the 4/3 cycle gates per Clifford is discussed, but comparing the quoted error with the coherence limit the numbers seem more in line with an individual beamsplitter gate. The authors should clarify this in the manuscript, and if the latter is indeed what they are quoting, it should be noted that given that the error within a cycle is not twirled, extracting the error of an individual gate from the cycle error could have significant systematic bias if the underlying error is coherent. This would likely impact the leakage post-selected results more significantly.”***

All fidelities and time constants quoted in the text are for individual gates from the G_{DR} gateset defined in equation 9, but we have edited some of the wording in main text lines 404 and 419-421 to help clarify this point. Additionally, the factor of 4/3 is not a conversion factor from our gateset to the Cliffords, but instead from our gateset to a single beamsplitter gate. This is because 4/6 of the gates in G_{DR} are themselves single beamsplitters, and the other 2/6 gates are swap operations that are implemented with two consecutive beamsplitters. So, to convert from an average single gate fidelity over the gateset G_{DR} to a single beamsplitter fidelity, we use a conversion factor of $4/6 \cdot 1 + 2/6 \cdot 2 = 4/3$ beamsplitters per average gate in G_{DR} . This calculation has been explicitly added to the end of supplementary section 9, hopefully making this much clearer.

As pointed out, this concatenation will indeed lead to increased sensitivity to coherent errors, as there is no twirling within the concatenation. This implies that the fidelities that we quote are lower bounds on the fidelities that would be extracted from a more conventional RB protocol as our scheme is more sensitive to coherent errors like miscalibration. We also agree that this effect would be more prevalent in the leakage-detected data, as even rather minor miscalibrations will contribute to infidelity when performed thousands of times.

This can be seen in our newly added supplementary figure S7a, where we compare a more conventional single-gate RB without concatenation to an RB curve using 5x concatenated gates. The two curves were taken on the same day and are from the same cooldown as all other RB data shown in the paper, and the single-gate RB only goes up to the aforementioned ~ 950 gate FPGA limit. Notably, the leakage-detected single-gate curve decays at a slightly slower rate than the 5x leakage-detected data. This signifies that the lack of twirling within this concatenation has allowed some coherent errors to accumulate more than they otherwise would have in a

non-concatenated sequence. The fact that we can still achieve a leakage-detected fidelity of 99.98% even with a 9x concatenation is a testament to the precision and accuracy of our pulse calibration technique detailed in supplementary section 8.

- 4. “Why did the authors not perform direct RB with a gateset composed of single beamsplitter gates? Did the authors try and it resulted in inconsistent results or non-exponential decay (perhaps due to leakage accumulation between gates)? I believe the manuscript would be improved by some further discussion of the reasoning for this non-standard benchmarking approach, perhaps in the methods or supplementary information.”***

As mentioned above, please see the newly added supplementary figure S7a for direct RB with a gateset composed of single beamsplitter gates and swap operations (which are implemented by two beamsplitters concatenated together, as described in main text figure 3e).

We find that this single-gate direct RB does indeed follow a smooth exponential decay with no inconsistent or unexpected results. However, we are limited from extending this exact curve to longer sequence lengths by the FPGA, but we cross-check our concatenation method by comparing this direct RB sequence to a 5x concatenation sequence to find that they are consistent with each other.

- 5. “One minor comment is that it would be useful to also plot error bars over or the sequence values of individual seeds, not just the mean of the RB decay curve. Showing the spread of individual sequences is useful for qualitatively assessing the coherence of the underlying noise process.”***

We have updated the figure legends to more explicitly show that all data points on RB plots have error bars, even if they are so small as to be hidden by the marker.

On the point of error bars for individual seeds, we would like to clarify our sampling protocol. Rather than generating ~ 100 random circuits, sampling each circuit $\sim 10^3$ times, and then averaging the averages, we are instead generating random circuits on a shot-by-shot basis. This means that all $\sim 10^5$ of our sequences are unique random circuits, sampled only once. Due to this, we don't think it is possible to plot error bars of individual seed circuits as each circuit is only sampled once.

Reviewer 2

The manuscript by Lu et al describes the experimental demonstration of a high-fidelity gate between two 3D microwave cavities. The authors developed novel experimental techniques that allows them to exploit the natural symmetries of a dc-squid and use it as a parity protected coupler. The work is novel, and the discussion is clear, both on the theory and experiment side. Overall, the novelty/impact of the work and the quality of the manuscript meet the requirements for publication in Nature Communication.

We thank the reviewer for evaluating our work as novel and for recommending for publication in Nature Communications. Their attention to detail helped bring forth some key points in the paper. Below we address point-by-point the comments raised by the reviewer.

- 1. "Line 36- 45: The deleterious effects of strong drive are also very much noticeable in amplifier type devices, so the statement seems a bit arbitrary. For example, "unwanted interaction with numerous [...] modes" is definitely an issue in other parametric systems, even low Q ones."***

We agree with this comment, unwanted interactions with parasitic modes are definitely an issue in amplifier-type devices too. When trying to engineer a high-fidelity parametric process, these undesired interactions need to be suppressed to rates lower than the relevant linewidths in the system. We believe that in our system, where a high Q converter is directly capacitively coupled to cavities with even higher Q, even processes that are much weaker than the converter linewidth might appear as a significant roadblock to achieving a process limited only by single-photon loss of the cavities, requiring an even higher standard of sanitation. We have changed the introduction (see lines 59-72) to stress further on the high-Q nature of our application, and have removed the comparison to amplification and traveling-wave applications.

- 2. "Line 54-67: It is unclear at this point how the high Q resonator participate in θ_c . I would move here Eq 3 and associated text to help the discussion."***

We have changed the introduction to no longer refer to the cavity participation in this initial discussion of the Hamiltonian's symmetry. Instead, the cavities are only brought in in Results Section 2 (Demonstrating a high-coherence beamsplitter), where their participation is used to derive the beam-splitting Hamiltonian, thus maintaining a coherent story.

- 3. "Line 96: θ_1 and θ_2 are not defined."***

We have now defined them in line 121.

4. ***“Line 139-161: This paragraph is kind of obscure without theoretical support. Also, not sure that a “squeezing resonance” is a thing. I would rephrase that this section.”***

We agree that it was difficult to parse through the theoretical discussion in the original manuscript without support from equations and numerics. We have moved this section to Supplementary Section 2, where it has been merged with related simulations and equations, which we hope now form a more coherent story. We have also clarified that the 'squeezing resonance' implies a resonant two-photon process at a drive frequency equal to half the converter g-f transition frequency, and given intuitive explanations for why this could lead to heating (see Supplement lines 155-162).

5. ***“Line 293-297: why not GRAPE? Is there a trade-off in efficiency, robustness, simplicity?”***

This is a great question - fast, deterministic, and high-fidelity preparation of the single photon cavity state is an important requirement in dual-rail protocols, particularly for converting detected leakage errors into a reset qubit (erasure conversion [1, 2]). In our manuscript, this preparation was not fully optimized, since our primary task was to characterize beamsplitter fidelity, which can be extracted through randomized benchmarking in a way that is reasonably insensitive to SPAM errors. We used the displacement + readout technique as a simple protocol whose fidelity was primarily limited by the fidelity of the ancilla-based dispersive readout. Since the time taken for this readout is not significantly longer than the speed limit placed by the cross-Kerr between the cavity (Bob) and ancilla, we obtain similar fidelities to naïve GRAPE-based preparation. In the future, more complicated protocols may be implemented, including resonantly swapping excitation from the ancilla to the cavity.

[1] Y. Wu *et al.* Nat Commun 13, 4657 (2022)

[2] J.D. Teoh *et al.* ArXiv:2212.12077 (2022)

6. ***“Line 322-323: why are you only fitting short sections and not the whole trace? Are there drifts?”***

The drifts in our calibration controls are negligible on the timescale of acquisition for these experiments. A better characterization of the magnitudes of these drifts can be found in the last paragraph of Supplementary Section 8. Choosing to fit short sections is only because acquiring the entire trace (eg Fig 2b) takes significant time at each amplitude point, and contains redundant information.

7. ***“Line 329: giving an order of magnitude of the shift could be useful here (it is currently buried in SI)”***

We agree, and have included an approximate shift (200 MHz) in the main text in line 287.

8. ***“Line 345: the word ‘implies’ seems too strong. Here you use measured quantities to estimate an upper bound for the fidelity.”***

We agree, and have changed lines 300-301 to clearly state that it's the decoherence-based upper bound on the fidelity.

9. ***“Line 353-357: at this point it is unclear that the coupler has its own drive and readout line. Is the coupler prepared in g via a driven reset protocol as well? Would a drive and readout line be present on the coupler line in the future, or is it mainly here to investigate how well it works?”***

This is an excellent point. We do indeed have a drive and readout on the coupler, and have changed lines 218-221 to explicitly mention this. The coupler is indeed prepared in g at the start of each of these experiments, and the readout is used to post-select on the coupler remaining in the ground state in the RB protocol detailed in Fig 3b. In our experiments, it was important to include this readout to characterize the immensely suppressed heating of the coupler, which was a primary claim in the manuscript. In future experiments, the readout may primarily be necessary only to ensure the coupler is at the flux sweet-spot, but alternative schemes for reading out the coupler through its dispersive coupling to the storage cavities are also possible, since this characterization can be slow and does not need to be performed too often.

10. ***“Line 364: what are the units of the heating rate?”***

The units for the heating rate are excitation [unitless] per swap. We have modified line 321 to make this clearer. We believe these heating units to be better from a practical standpoint, than the equivalent rate of $2\pi * 40$ Hz.

11. ***“Line 408: If I understand well, that means that now you post-select on the coupler being in the ground state, the ancilla being in the ground state, cavity photon loss, and number resolved measurement to prepare $|0a1b\rangle$? Overall, what is the fraction of runs post-selected out?”***

The selection on these conditions can be separated into preparation and post-selection. Our preparation is done through a "proceed only if you succeed" policy. For example, the preparation of $|0a1b\rangle$ is done by displacing Bob to a coherent state ($\alpha = \sqrt{2}$) and using the dispersive with the ancilla to selectively flip the ancilla only if Bob is in Fock 1. This preparation step proceeds only if the ancilla is indeed found to be excited (or in the ground state after another un-selective pi flip), and if not, it lets the cavity thermalize and tries again. This step thus has a success probability corresponding to the product of the ancilla readout fidelity ($\sim 82\%$) and the overlap of Fock 1 with the displaced state ($\sim 27\%$), which gives a total of $\sim 22\%$. We stress that this is a preparation step and does not affect the gate fidelity in any way.

For the post-selection protocols, the post-selection on no cavity loss (leakage) succeeds with a probability exceeding 99.9% per gate, which means we need to discard only one out of every 1390 shots per gate to boost our gate fidelity to 99.98%. For the longest sequences considered (2250 gates), this means we discard 80% of the shots on leakage detection.

The post-selection on ancilla heating is primarily done in this experiment to discount first-order effects of the ancilla and focus on coupler-induced dephasing, and may not be a necessary part of the protocol in future experiments. This discards shots with a probability $\sim 10\%$, dictated primarily by the thermal population of the ancilla.

Finally, the post-selection on the coupler state is only to demonstrate that the RB curve shows essentially no change on selecting out events in which the coupler remained in g , reinforcing our point of this beamsplitter not being limited by coupler-induced decoherence even at strong beamsplitter strengths. This discards shots with a probability $\sim 1.5\%$, again dictated by the coupler's residual thermal population.

12. “Line 432-449: the improvement under error-detection is useful if you post-select, but can it be error corrected?”

This is an excellent question and has been the primary focus of the rapidly growing field of erasure conversion. While cavity decay projects the qubit onto an orthogonal detectable state, this projection does not obey the Knill-Laflamme conditions and requires concatenation with a larger code to fully error correct. In particular, Teoh et al [1] show that these detectable leakage errors essentially scale the same way as first-order error correction, when integrated into a regular surface code. This however relies on the assumption that there exists a strong error hierarchy between the cavity decay and dephasing (dephasing much smaller than decay), which in our manuscript we refer to as a ‘noise-bias’. Even in regular photon-loss correcting bosonic codes [2, 3] (not necessarily single-photon), preserving this noise bias while engineering a strong multi-mode interaction is crucial. This is why we use our post-selection protocols to stress the suppression of coupler-induced cavity dephasing, and show that the cavities are still noise biased (approximately a factor of 4 in this implementation, limited primarily by a hot transmon ancilla). Improving this number is a subject for future work.

[1] J.D. Teoh *et al.* [ArXiv:2212.12077](https://arxiv.org/abs/2212.12077) (2022)

[2] N. Ofek *et al.* [Nature 536, 441–445](https://doi.org/10.1038/nature20854) (2016)

[3] V. V. Sivak *et al.* [Nature 616, 50–55](https://doi.org/10.1038/s41586-023-03000-0) (2023)

13. “Section 4 in the SI: it would be useful to make a brief mention in the main text that the DC flux offset is calibrated out.”

We have changed line 148 in the main text to explicitly mention this.

14. ***“Figure S5: Could you label what the different driving pins are for (ancilla/coupler drive/readout, etc)”***

We have now labeled all the drive and readout pins explicitly (see figure S8 in the revised manuscript)

15. ***“One overall comment is that the manuscript switches from theory to experimental/setup description a few times, making the manuscript a bit hard to follow. It would really help to finish the theory part before moving to the physical implementation.”***

We agree, the intertwined explanations of novel theory and experimental techniques were difficult to navigate in the original manuscript. This motivated much of the reformatting of the manuscript to only make some intuitive commentary on the theory in the introduction, leaving the details for Supplementary Section 2. The rest of the sections then primarily focus on experimental implementation and results.

Reviewer 3

This article presents a high-fidelity beamsplitter coupling between two harmonic oscillator modes (3D EM-cavities), which can be turned on and off as it results from parametric driving. The process is achieved through the well-known Josephson junction nonlinearity, and a symmetric squid-architecture is used to select only even parity terms. The precision obtained on the 01-10 Fock space with this scheme is indeed very high.

I have two main points of criticism on the manuscript in its present form.

We thank the reviewer for their incisive comments - we believe that answering these two main points of criticism significantly helped improve the clarity and impact of our manuscript, including encouraging us to carry out a detailed analysis of the limitations from coherent drive-induced nonlinearity and to better emphasize the importance of our post-selection protocols. Please see our detailed responses below.

- 1. “First, I somewhat fail to see the originality of the scheme compared to previous realizations like [15,33] cited by the authors. The main text remains rather vague, and the reference to supp. mat. is not focused enough. Therefore, it is not clear enough which main novel idea leads to this performance — or is it just the 3D cavities, plus restricting to 0,1 Fock space?”***

We thank the reviewer for the comment, and we regret the lack of clarity in our initial draft. To improve the structure of the manuscript and better match the format of *Nature Communications*, we have rewritten the introduction with more direct and concise explanations. On the point of originality and how this work should be considered in light of the JPC and ATS [15 and 33 in the original manuscript], please see our revised lines 59 through 72 of the main text. These prior works do indeed utilize multi-mode circuits to engineer a parity symmetry for cleaner parametric processes, and we build off of these same ideas with the SQUID. Specifically, the originality of our work lies in the extension of these symmetry ideas to a high-Q environment where even minor imperfections in converter design can spoil the high coherence of the cavity storage modes. To highlight an example of some of the difficulties of working in this regime, we point out that in the initial implementation of the ATS [1], the on-chip resonator that is storing the cat state has a T1 of 3us, limited by unintentional purcell decay through the flux delivery lines for the ATS. In contrast our implementation operates with quality factors $\sim 100\times$ larger with storage mode T1s of 300us. This required a much more detailed understanding of the interaction of the cavity with both the coupler and control line, and represents a significant improvement in the state of the art for high-performance parametric operations.

To clarify whether the improvements demonstrated in this work are primarily due to using 3D cavities and restricting to the single photon subspace, we would like to compare and contrast our results with prior work [2] involving a single-junction transmon converter. This work employs the same 3D post-cavity designs used in our system, with $\sim 30\%$ better cavity coherence times.

Yet, when extracting a decoherence limit on their beamsplitter fidelity within the same single-photon subspace, they find an infidelity of $\sim 1\%$ as compared to our 0.04% , which means that we have over an order of magnitude improvement even with our slightly lower-lifetime cavities.

This significant improvement was only made possible because other important roadblocks to the fidelity, like coupler-induced dephasing and coherent parasitic nonlinear processes, were removed by identifying and fine-tuning useful symmetries. In particular, we see an enormous reduction of drive-induced coupler heating from $\sim 2\%$ excitation per swap in [2], to $4\text{e-}3\%$ excitation per swap in our work. This is an important change, since the dispersive shift to the coupler can change the resonance condition for beamsplitting by an amount comparable to or greater than the strength of the beamsplitter, when the coupler is excited. The beamsplitting strength g_{BS} can also directly depend on the driven coupler state [3]. The resulting improvements by more than 100x in speed and fidelity of the beamsplitter reported here are enabling and inspiring new approaches to quantum computation and error correction [4,5].

To bring home these points better, we have restructured our introduction to stress on undesired driven effects as the primary roadblock, which we combat using similar symmetry principles as in the ATS and JPC, but now applied to high-Q. All details of the theoretical discussion have been moved to and directly integrated with the Supplement, in particular Supplementary Section 2. We have also added short phrases to better communicate our references to the supplement, which contain many details that we believe will be extremely useful to someone attempting to replicate this result. We thank the reviewer for pointing out the weaknesses in our original structure, and we believe the revised introduction has greatly simplified the main messages of the manuscript, and has improved its readability and impactfulness in general.

[1] R. Lescanne *et al.* Nat. Phys. **16** 509-513 (2020)

[2] Y. Gao *et al.* PRX **8**, 021073 (2018)

[3] Y. Y. Zhang *et al.* Phys. Rev. A **99**, 012314 (2019)

[4] J. D. Teoh *et al.* ArXiv:2212.12077 (2022)

[5] T. Tsunoda *et al.* ArXiv:2212.11196 (2022)

- 2. "Second, I indeed find quite reductive to restrict the analysis to this 0,1 Fock space. While this may be one choice for qubit encoding, it is certainly not the only one, and it is certainly not the only inputs expected into a "beamsplitter". At higher Fock numbers, we may expect that nonlinearities and higher-order processes could significantly alter the functioning of the device, so probing 0,1 Fock is really not convincing evidence for claiming the announced beamsplitter Hamiltonian. In fact, for some readers it may even be quite suspicious to have no extensive comments on this."***

We fully agree with the reviewer that, while residual nonlinearities and higher-order processes don't play an important role in our experiment which aimed to demonstrate high-fidelity dual-rail qubit operations, they could induce coherent errors for high-photon-number schemes. To address this concern, we have added a new paragraph (see lines 331-347) in the main text to clarify the potential limitations due to these nonlinearities, and the future steps that can be taken to mitigate them. To expand on this discussion, we have also added a new supplementary section (Supplementary Section 6). In this section, we have included additional data and discussion showing experimental and simulation results on using this device with larger photon numbers. These simulations demonstrate that by implementing modifications compatible with our design, we can achieve a substantial improvement in performance.

To elaborate further, our SQUID coupler is in the transmon regime and is dispersively coupled to the storage cavities. Inevitably, the cavities will inherit a small amount of nonlinearity from the coupler. We measured the total cavity Kerrs to be around 5~15 kHz when the coupler is undriven (see supplementary section 12). A significant contribution to this total static Kerr comes from the dispersively coupled transmon ancilla, which bounds the minimum nonlinearity of the system even for a perfectly Kerr-free coupler.

In our frequency configuration, when the coupler is driven, the Zeeman shift (~200 MHz) brings the coupler and its sidebands closer to the cavities, due to which we measure a marked increase in the cavity Kerrs (see supplementary section 6). Such a combination of sideband collision and frequency shift is an issue for 4-wave-mixing couplers (including the charge-driven transmon cite!) in general, and is not unique to our differentially-driven SQUID. At our operating point where the beamsplitter rate is around 2.2MHz, we measure a drive-induced cavity Kerr around 128kHz (for Bob; Alice is estimated to be around half this value). We believe, however, that this is a solvable problem, and there exist multiple solutions that are quite compatible with our architecture.

The simplest possible improvement would be to change our coupler frequency to be further detuned from the cavities. This would result in a delayed sideband collision point in our experiment, which would increase the g_{bs}/Kerr ratio at the operating point (and in the same vein helpful for improving the g_{bs}/loss ratio and the overall beamsplitter fidelity). We show through Floquet simulations (see Fig. S4a) that increasing the coupler frequency by just 0.5 GHz would suppress the drive-induced Kerr by a factor of 6.

Additionally, we could redesign the coupler as an array of dc SQUIDs, which greatly suppresses the static Kerrs by roughly a factor of N^2 , where N is the number of the SQUID loops. We verify in the numerical simulation that for our specific system, even with an array of only $N = 3$ SQUIDs, the drive-induced Kerr would be suppressed by a factor of 15. Further, by combining this arraying with a shift to higher frequency we could achieve a total suppression by more than a factor of 50.

Finally, during the beamsplitter pulses, the cavity Kerr can also be dynamically cancelled out through additional microwave drives on the coupler, as shown in [1]. Kerr can also be corrected via additional unitaries like SNAP [2], which is already required by most applications to control bosonic modes in a high-fidelity manner.

Each of these solutions have been simulated and discussed in the additional Supplemental section (Section 6).

Finally, to place these quantities in context and better understand their implications on gate fidelity, we refer the reviewer to the newly added Fig. S4b in the supplement. Here, we perform time-domain simulations of a swap pulse being applied to various coherent states in the presence of both static and drive-induced Kerr, computing the state fidelity with the ideal swapped coherent state, and thus finding the infidelity limit due to this coherent error. We find that by implementing a higher frequency, arrayed coupler we would be able to swap coherent states of size $\alpha = 2$ with Kerr-limited fidelities exceeding 99.9%, and over 99% for $\alpha = 3$.

While it is true that residual nonlinearity will always affect performance for states spanning a broad range of photons, significant progress towards realizing hardware-efficient error correction has been made with protocols using states with photon number distributions similar to the states simulated here [3,4,5]. If this driven Kerr is indeed a substantial issue, then exploring our proposed solutions or an additional final unitary (like in [2]) will be required.

- [1] Y. Zhang et al, PRA 105, 022423 (2022)
- [2] R.W. Heeres et al. PRL 115, 137002 (2015)
- [3] N. Ofek et al, Nature 536, 7617 (2016)
- [4] J. M. Gertler et al, Nature 590, 243 (2021)
- [5] Z. Ni et al, Nature 616, 56 (2023)

- 3. *“Finally, as a minor comment, I would drop the post-selection part, or at least displace it to sup.mat only. Indeed, post-selection-based schemes always come with a grain of salt, so when this implies no too significant gain — like seems to be the case here — it is not helping your case to insist too much on this variant.”***

We thank the reviewer for this comment. We believe that the dramatic gain on post-selection - a reduction of the error rate from 0.08% to 0.02% - is a major result for several reasons. First, this can allow major improvements for shallow circuits via post-selection, and can be used in more complex schemes like concatenation into a surface code (see [1]). To express this in a different way, by discarding only one out of every 1390 (0.07%) shots per gate, the gate fidelity improves by almost a factor of 4. Specifically, the number of shots to throw out after a sequence of N

gates scales exponentially ($\sim e^{-N/1390}$), and even for the longest sequences considered here (2250 gates), we still keep 20% of the shots after leakage detection.

Second, this improvement by a factor of 4 directly provides a quantification for how well we have been able to preserve the intrinsic error hierarchy between decay and dephasing of the 3D cavities. Being primarily dominated by single-photon loss in the cavities, which is a correctable error for various bosonic codes [2-7] allows gates constructed from this beamsplitter to be compatible with the gains from those schemes. Our additional comparison to the post-selection on the coupler state shows that this performance is not limited by coupler-induced dephasing, and thus the noise-bias may be further improved by managing the remaining dephasing environment of the cavities, for example from hot ancillary qubits.

Thanks to the reviewer pointing this out, we have amended our manuscript to emphasize the above points, and we hope this makes the improvement clear to the readers. The significant improvement on postselection of detectable leakage, proving the preservation of the noise-bias, has been highlighted in lines 432-439. The postselection on the coupler has been emphasized in lines 406-411.

- [1] J. D. Teoh *et al*, ArXiv:2212.12077 (2022)
- [2] I. L. Chuang *et al*, PRA 56, 1114 (1997)
- [3] P. T. Cochrane *et al*, PRA 59, 2631 (1999)
- [4] Z. Leghtas *et al*, PRL 111, 120501 (2013)
- [5] N. Ofek *et al*, Nature 536, 7617 (2016)
- [6] J. M. Gertler *et al*, Nature 590, 243 (2021)
- [7] Z. Ni *et al*, Nature 616, 56 (2023)

Summary of Major Changes

We summarize here the changes to each section of the originally submitted (henceforth called the 'old') manuscript, with reference to the revised ('new') manuscript.

1. Main text

- a. **Introduction:** Revised to shorten the theoretical discussion, with the comparison to the transmon Hamiltonian being moved to Supplementary Section 2. More stress has been placed on the idea of converter symmetry being an established useful resource, with this work applying it to a high-Q environment.
- b. **Old Section 1 (Advantages of the differentially-driven SQUID):**
This section has been entirely absorbed into other parts of the paper. The basic ideas of utilizing the symmetry in the SQUID Hamiltonian have been moved to the new Results Section 1, whereas the more intricate details of why this symmetry offers an advantage have been moved to Supplementary Section 2.
- c. **Old Section 2 (Engineering a highly differential drive):**
Now the new Results Section 1, with minor changes to integrate it better with the introduction.
- d. **Old Section 3 (Demonstrating a highly coherent beamsplitter):**
Now the new Results Section 2, with a few sentences added to address driven nonlinearity.
- e. **Old Section 4 (Benchmarking beamsplitter performance):**
Now the new Results Section 3, with minor changes to address Reviewer 1's comments.
- f. **Old figure 3:**
Added error bars to all RB plots
- g. **Old Section 5 (Outlook):**
Now the new Discussion section, with a sentence added to address driven nonlinearity

2. Methods

No major text changes, but some sections have been moved to the Supplement to better fit the format requirements or improve readability:

- a. Old Sections 6.1 and 6.2 have been exchanged
- b. Old Sections 6.5 and 6.6 are now new Supplementary Sections 5 and 8 respectively

3. Extended Figures

- a. Old Extended data Figure 1 is now new Fig. 5
- b. Old Extended data Figure 2 is now split into new Figures 4, 6, and 7 respectively. New Fig. 4 now has an example fit for the chevron pattern.
- c. Old Extended data Figure 3 is now new Supplementary Fig. S3
- d. Old Extended data Figure 4 is now new Supplementary Fig. S6
- e. Figure captions have also been modified to better fit the context of the surrounding text

- f. White space has been trimmed to be more compatible with inline figures

4. Supplementary Information

- a. Sections have been rearranged in the following way:

Old	New
Section II	Section 12
Sections IV, V	Sections 3, 4 respectively
Sections VI and VII	Merged into Section 7
Sections VIII, IX, X, XI	Sections 9, 10, 11, 13 respectively

- b. Old Section 3 is now new Section 2. There have been major additions in text to include old Main text Section 1. This now singularly serves as the section addressing theoretical explanations of the advantages of the differentially-driven SQUID.
- c. New Section 6 and Fig. S4 have been added to directly address reviewer 3's concerns about driven nonlinearity
- d. Old Fig. S4 has been modified into new Fig. S7 by including a plot that shows that repeating gates in the RB sequence still provides a lower bound on gate fidelity
- e. Old Fig. S5 has been modified into new Fig. S8 with all drive and readout pins now labelled

REVIEWERS' COMMENTS

Reviewer #1 (Remarks to the Author):

I would like to thank the authors for their thorough consideration of my comments, and their detailed responses. I believe all my questions have been addressed in a satisfactory way by the response letter and changes to the manuscript. I maintain my strong recommendation for publication in Nature Communications.

Reviewer #2 (Remarks to the Author):

Thank you for your detailed answers. I believe all the points raised by myself and the other referees have been properly addressed, and the manuscript is now suitable for publication in Nature Communication.

Reviewer #3 (Remarks to the Author):

The additional clarifications made by the authors satisfactorily answer my two main points. I have always been convinced that the device, as such, was a major achievement, and now that its main framework is more explicit, I believe that it is indeed publishable in Nature Communications.

There is a typo regarding my minor point, in the reply-to-review: you are not throwing out a number or fraction $\exp(-N/1390)$ shots for a sequence of N gates, but you are rather keeping a fraction $(1-1/1390)^N$. The 20% conclusion for 2250 gates is correct though.

Whether this is acceptable and whether an error-rate decrease by a factor 4 in this context is "dramatic" as the authors claim, is probably context-dependent, and much more open to debate than the device performance itself.

When error-detection allows to pick out photon-loss at rate $1/1390$ per gate cycle, we can repeat exactly the same argument and conclude that error correction would essentially not be necessary because post-selection is great. But that's not how we see it. The large- N scaling is bad, so that's not the solution.

Error-detection and post-selection are always an option, with specific numbers governing their efficiency; and so i would find it weird to include it as a whole section of the main paper when there is no truly new principle with it. For me, it rather distracts from the main message, and thereby weakens it.

Dear Referees,

Thank you for your kind comments and for recommending publication in Nature Communications. We believe your detailed and incisive review greatly helped improve the clarity of this manuscript and polish key details.

Referee 1

I would like to thank the authors for their thorough consideration of my comments, and their detailed responses. I believe all my questions have been addressed in a satisfactory way by the response letter and changes to the manuscript. I maintain my strong recommendation for publication in Nature Communications.

We thank the referee for their strong recommendation for publication and their positive impact on our manuscript.

Referee 2

Thank you for your detailed answers. I believe all the points raised by myself and the other referees have been properly addressed, and the manuscript is now suitable for publication in Nature Communication.

We thank the referee for their recommendation for publication, and their very detailed review.

Referee 3

The additional clarifications made by the authors satisfactorily answer my two main points. I have always been convinced that the device, as such, was a major achievement, and now that its main framework is more explicit, I believe that it is indeed publishable in Nature Communications.

There is a typo regarding my minor point, in the reply-to-review: you are not throwing out a number or fraction $\exp(-N/1390)$ shots for a sequence of N gates, but you are rather keeping a fraction $(1-1/1390)^N$. The 20% conclusion for 2250 gates is correct though. Whether this is acceptable and whether an error-rate decrease by a factor 4 in this context is "dramatic" as the authors claim, is probably context-dependent, and much more open to debate than the device performance itself.

When error-detection allows to pick out photon-loss at rate 1/1390 per gate cycle, we can repeat exactly the same argument and conclude that error correction would essentially not be necessary because post-selection is great. But that's not how we see it. The large-N scaling is bad, so that's not the solution.

Error-detection and post-selection are always an option, with specific numbers governing their efficiency; and so i would find it weird to include it as a whole section of the main paper when there is no truly new principle with it. For me, it rather distracts from the main message, and thereby weakens it.

We thank the referee for their comments, we share their perspective of the experimental results being a major achievement and therefore highly appreciate their critique in polishing our theoretical arguments.

For the minor comment, we indeed made a typo in our response (but not in the manuscript) - the number we quoted to be exponentially decreasing is the number of shots we keep. Our quoted ' $\exp(-N/1390)$ ' formula was an estimate from an exponential fit to the curve obtained from measuring the total number of shots that remain in the single-photon subspace as a function of the number of RB gates (see Supplementary figure). This is slightly different from the assumption of a constant probability of failure per shot, which we believe would match the referee's predicted formula, but nonetheless, the two formulae agree to significantly more than our experimental precision.

We indeed agree that the large-N scaling for postselection is bad. However, our aim in this manuscript was not to suggest post-selection as a route towards a quantum circuit architecture. Instead, it was to show the remaining errors after selecting out single-photon loss in the cavities is significantly lower than the leakage rate. This preservation of noise-bias for long-lived superconducting resonators is important in applications like bosonic encodings that correct amplitude loss and concatenations with surface codes that benefit from this error hierarchy [1]. We have therefore decided to keep this section in the main text, and are hopeful that this will inform future applications of our noise-bias-preserving beamsplitter. Thank you again!

[1] J.D. Teoh *et al.* [ArXiv:2212.12077](https://arxiv.org/abs/2212.12077) (2022)